# An atypical EhGEF regulates phagocytosis in *Entamoeba histolytica* through EhRho1

**Ravi Bharadwaj**[1], **Tushar Kushwaha**[2], **Azhar Ahmad**[3], **Krishna K. Inampudi**[2], **Tomoyoshi Nozaki**[4]*, **Somlata**[3]*

1 Division of Infectious Diseases, Department of Medicine, University of Massachusetts Medical School, Worcester, Massachusetts, United States of America, 2 Department of Biophysics, All India Institute of Medical Sciences, New Delhi, India, 3 Multidisciplinary Centre for Advanced Research and Studies, Jamia Millia Islamia, New Delhi, India, 4 Department of Biomedical Chemistry, Graduate School of Medicine, The University of Tokyo, Tokyo, Japan

* nozaki@m.u-tokyo.ac.jp (TN); somlata83@gmail.com, somlata@jmi.ac.in (S)

**Data Availability Statement:** All relevant data are within the manuscript and its Supporting Information files.

## Abstract

The parasite *Entamoeba histolytica* is the etiological agent of amoebiasis, a major cause of morbidity and mortality due to parasitic diseases in developing countries. Phagocytosis is an essential mode of obtaining nutrition and has been associated with the virulence behaviour of *E. histolytica*. Signalling pathways involved in activation of cytoskeletal dynamics required for phagocytosis remains to be elucidated in this parasite. Our group has been studying initiation of phagocytosis and formation of phagosomes in *E. histolytica* and have described some of the molecules that play key roles in the process. Here we showed the involvement of non-Dbl Rho Guanine Nucleotide Exchange Factor, EhGEF in regulation of amoebic phagocytosis by regulating activation of EhRho1. EhGEF was found in the phagocytic cups during the progression of cups, until closure of phagosomes, but not in the phagosomes themselves. Our observation from imaging, pull down experiments and down regulating expression of different molecules suggest that EhGEF interacts with EhRho1 and it is required during initiation of phagocytosis and phagosome formation. Also, biophysical, and computational analysis reveals that EhGEF mediates GTP exchange on EhRho1 via an unconventional pathway. In conclusion, we describe a non-Dbl EhGEF of EhRho1 which is involved in endocytic processes of *E. histolytica*.

## Author summary

*E. histolytica* is causative agent of amoebiasis in humans, which is of major concern in children, as repeated infection leads to stunted physical and mental growth. Parasite display variety of endocytic processes, like phagocytosis, trogocytosis and micropinocytosis to name a few. The molecular mechanism of these processes is still largely unknown. The molecules, which participate in these endocytic processes may prove to be good therapeutic targets, as endocytic processes are necessary for parasite growth, survival, and pathogenesis in host system. Here in this report, we have focussed on an unconventional EhGEF, which was identified as EhRho1 interacting protein by mass spectrometry. Our

**Funding:** The study was funded by Department of Science and Technology- SERB division by ECR/2018/000971, WEA/2020/000010 grants and UGC India by Startup Grant awarded to Somlata. Grant-in-Aid for Challenging Research (Exploratory), JP21K19372 to T.N. from Japan Society for Promotion of Sciences (JSPS), and Grant for research on emerging and re-emerging infectious diseases from Japan Agency for Medical Research and Development (AMED, JP20fk0108138 to T.N.). The funders had no role in study design, data collection and analysis, decision to publish, or preparation of the manuscript.

**Competing interests:** The authors have declared that no competing interests exist.

microscopy experiments show EhGEF participates in phagocytosis and trogocytosis and is highly dynamic in nature. The molecule also plays a crucial role in activation and recruitment of EhRho1 to the site of phagocytosis, and downregulation of EhGEF mimics phenotype similar to overexpression of activity dead mutant (T34N) of EhRho1. Our finding reveals EhGEF to play important role in actin dependent endocytic processes through EhRho1 and thereby, regulating actin dynamics during the process through EhFormin1 and EhProfilin1.

## Introduction

*Entamoeba histolytica* is the causative agent of amoebic dysentery or amoebiasis, a major public health problem throughout the world, particularly in developing countries and is one of the major causes of morbidity and mortality [1,2]. The parasite can invade both intestinal as well as extra intestinal sites and in the absence of proper diagnosis or treatment, it can be fatal [3,4]. *E. histolytica* is a highly phagocytic cell, and phagocytosis plays a key role in amoebic pathogenesis. Phagocytosis is also an essential route of nutrient uptake in *E. histolytica* as blocking this process leads to inhibit the cell proliferation and pathogenicity [5,6]. However, the mechanism of phagocytosis, particularly the initial steps leading to phagocytic cup formation up to phagosome closure in *E. histolytica*, is not clearly understood unlike metazoan systems where the process has been studied in extensive detail [7,8].

Phagocytosis is an important process in most eukaryotic systems and it is involved in many functions, including clearance of pathogens and necrotic or apoptotic cells [9]. Phagocytosis in *E. histolytica* is likely to involve unique molecules compared to mammals as a number of molecules known to be involved in mammalian phagocytosis could not be identified in this organism[10]. A number of cell surface molecules, such as Gal/GalNAc lectin [11], TMK96 [12], TMK39 [13], SREHP [14,15] and EhROM1[16] have been shown to be involved in adherence to other cells. It is not yet clear if these molecules are amoebic receptors during phagocytosis of prey, such as RBC, bacteria and apoptotic human cells [17,18]. The participation of Gal/GalNAc lectin as a receptor in phagocytosis has been questioned, though it is likely that it may still be a key molecule initiating signal transduction [14,15,19]. Analysis of the phagosome proteome has revealed involvement of a large number of proteins in phagosome formation and subsequent maturation [19–22]. Some of these, such as actin, Arp proteins, actin binding proteins, PI3 kinase, activated protein kinase (PAK), and Rho GTPases are already known to be part of phagocytic and signalling pathways [23–27]. Though many of the identified molecules are suggested to be part of the phagocytic pathways, detailed molecular mechanisms have not yet been elucidated in this parasite. Rho family GTPases regulate many cellular signalling processes including actin cytoskeleton as one of them. *E. histolytica* genome codes for large number of small GTPases belonging to Rho, Ras and Rac families and it has been suggested that the function and regulation of these molecules are intimately related to pathogenic mechanisms of *E. histolytica* [23,28,29].

Interestingly, *E. histolytica* genome encodes more than 29 Rho domain containing GTPases as compared to 18 found in human [30,31]. The functional assignment of amoebic Rho proteins has not been carried out systematically, and only limited information is available regarding their role in amoebic biology. Particularly, we do not have much information about the participation of Rho GTPases in regulation of actin cytoskeleton in *E. histolytica*. Overexpression of a constitutively active mutant of EhRacG in *E. histolytica* altered cytokinesis and cell polarity due to concentration of F-actin at one end of the cell [32]. EhRacA and its effector

EhPAK2 are likely to be involved in regulation of cytoskeleton as delayed cytokinesis and a defect in phagocytosis were observed on over expression of EhRacA [23]. Though EhRho1 shows 47% sequence identity and 70% structural similarity with its human homologue, HsRhoA is distinctly different as it lacks the signature "Rho insert" domain that separates Rho proteins from Ras superfamily. Moreover, EhRho1 is not a substrate for *Clostridium botulinum* C3 exoenzyme, a critical feature of mammalian Rho GTPases, but is glucosylated by *Clostridum difficile* toxin B and *Clostridium novyi* α-toxin [33,34]. Several HsRhoA effector molecules have been reported, and shown to participate in regulation of actin cytoskeleton, whereas EhRho1 has only been shown to interact with ROCK-2-like protein, EhFormin1 and EhProfilin1[35].

Rho GTPases and their downstream effectors have been known to regulate cytoskeleton reorganisation in various cellular processes such as cytokinesis, motility, and apoptosis [36–38]. Rho GTPases act as molecular switches between active GTP-bound and inactive GDP-bound states due to their intrinsic GTPase activity. Rho GTPases are regulated by GEFs, GTPase activating proteins (GAPs), and guanosine diphosphate dissociation inhibitors (GDI) [39]. Binding of GEF induces the dissociation of Rho-GDI complex leading to conformational changes required for exchange from GDP to GTP. Once activated, Rho proteins translocate to membrane and interact with various downstream effectors. Meanwhile, interaction of GAP stimulates intrinsic GTPase activity of Rho proteins that accelerate the hydrolysis of GTP to GDP. After hydrolysis of GTP, GDI binds and sequesters the membrane binding prenylation domain of Rho GTPases that brings Rho-GDI complex back to cytosol [40,41]. In contrast, GEFs serve as positive regulators of Rho family proteins by catalyzing the exchange of GDP for GTP, which results in increased levels of active GTP-bound Rho GTPases in cell. Most Rho GEF proteins belong to the Dbl (diffuse B-cell lymphoma transforming protein) family. However, some unconventional, non-Dbl RhoGEFs also exist [42]. RhoGEFs share approximately 300 homologous amino acids consisting of two tandem and functionally inter-related segments known as Dbl homology (DH) and pleckstrin homology (PH) domains[43,44]. The DH domain (∼200 amino acid residues) is involved in the catalytic reaction that stimulates GDP/GTP exchange through binding to the GTPase and induces a conformational change in the latter. The PH domain (∼100 amino acid residues) is located immediately C-terminal to the DH domain. It has been reported that the primary role of the PH domain is to anchor the protein to the membrane by interacting with phosphoinositides [44,45].

Although some of the pathways involved in actin dynamics during phagocytosis have been partially elucidated, like recruitment of Arp2/3 subunits via EhAK1 but Rho mediated actin remodelling is not very well understood in *E. histolytica*. In this report, we have demonstrated the role of an unconventional EhGEF in phagocytosis of *E. histolytica*. Our results show that EhGEF is recruited to site of phagocytosis and trogocytosis in response to phosphatidyl inositol 3,4,5 triphophate (PtdInsP3) transiently and also recruits EhRho1 to the site. Also, EhGEF activate EhRho1 by altering the nucleotide state via a different mechanism which possibly involves binding of GTP to EhGEF itself first and then loading it to EhRho1. It has been shown previously that activated Rho1 recruits both EhFormin1 and EhProfilin1 to the phagocytic cups, and this results in actin nucleation and polymerisation at the site. Although *E. histolytica* genome codes for 62 Dbl-domain proteins which may act redundantly for Rho proteins, but this redundancy may be even higher due to existence of unconventional GEFs as well. Since the phagocytic and trogocytic pathways both contribute to the pathogenesis and survival of amoeba trophozoites in host system, understanding role of these unique molecule may lead to drug targets specific for parasite, which is useful as drug tolerance against only drug of choice metronidazole has already been reported [46].

## Results

### Identification of EhGEF as EhRho1 binding protein

Our previous work has shown the importance of EhRho1 in amoebic pathogenesis. Signalling through EhRho1 after attachment of RBCs appears to be crucial for progression of phagocytic cups towards phagosomes formation [35]. In order to identify the molecules that may be involved in phagocytosis along with EhRho1, we carried out immunoprecipitation from amoebic lysates using EhRho1 specific antibody. Mass spectrometric analysis of eluted fractions resulted in several EhRho1 bound proteins. The proteins identified by mass spectrometry analysis included known EhRho1 interacting partners namely EhFormin1, EhProfilin1and EhGAPs. However, there were some novel proteins that come out with high confidence score and among them a novel Guanine Nucleotide Exchange Factor (EhGEF, EHI_008090) was selected for further study on the basis of high abundance of peptides in mass spectrometry results and presence of well conserved PH domain, which is known to bind phosphatidylinositol 3,4,5 triphosphate (PtdInsP3). It is well known that PtdInsP3 is generated during endocytic processes and mediates downstream signalling. The putative GEF domain in this proteins is poorly conserved and is not similar to Dbl-RhoGEFs, which classify it as a nonconventional GEF.

EhGEF was identified as EhRho1-binding protein in an affinity screen (S1 Table). In order to validate the interaction between EhGEF and EhRho1, GST-tagged EhRho1, GST-EhRho1-Q63L (constitutively active form of EhRho1) and GST-EhRho1-T34N (dominant negative form of EhRho1) were incubated with total lysate of trophozoites expressing HA-tag EhGEF. Glutathione-Sepharose was used to pull down the complex and the presence of EhGEF was determined by using a HA tag-specific antibody. As results shown in **Fig 1A**, only wild type GST-EhRho1 was able to interact with EhGEF. However, EhGEF could not be detected in the pull down material of Q63L-EhRho1 or T34N-EhRho1. The interaction between two molecules was further investigated by immunoprecipitation using EhRho1 and HA-tag specific antibody. The EhRho1 antibody precipitated EhRho1 along with EhGEF from the total cell lysate and vice versa result was also found in immunoprecipitation with anti-HA-tag antibody (**Fig 1B**). The results suggest an interaction between wild type EhRho1and EhGEF but the latter is not able to interact with any mutant form of EhRho1.

### EhGEF binds phosphoinositides (PI)

*In silico* analysis of EhGEF indicated that the protein is composed of a conserved PH domain at N-terminal followed by a stretch of amino acid which is putatively assigned GEF like. As PH domain is known to bind PI and especially, PtdIns(4,5)P2 and PtdIns(3,4,5)P3, the HA tagged EhGEF (HA-EhGEF) expressed in trophozoite was also investigated for its lipid binding ability. The total lysate of HA-EhGEF expressing trophozoites were incubated with PI lipid array and probed with antibodies specific to HA tag. The results showed that the HA-GEF was able to bind PtdIns(3,4,5)P3 and some cross reactivity was also observed for PtdIns(3,4)P2 which is well documented in literature for this domain (**Fig 1C**). The HA-tagged PH domain alone (ΔEhGEF) also showed the same pattern of PI binding (**Fig 1D**). The results indicate that PH domain of EhGEF is responsible for PtdIns(3,4,5)P3 binding in the plasma membrane and also localize the recruitment of EhRho1 to the site of phagocytosis.

### EhGEF can exchange GTP for GDP on EhRho1

As EhGEF was confirmed to be EhRho1 binding protein, we assayed its GTP exchange property in order to identify its role in the interaction. In general, GEFs regulate Rho GTPases

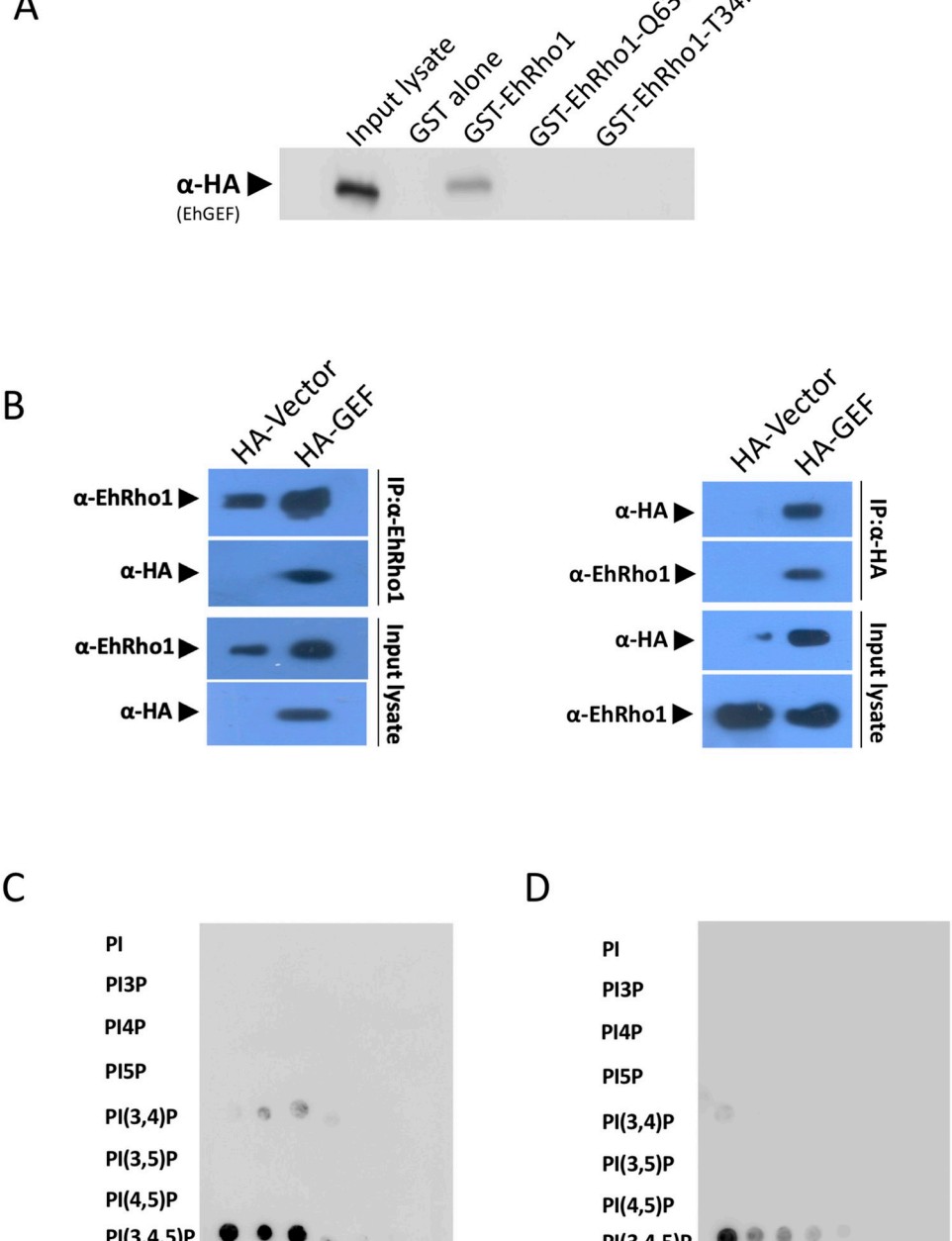

**Fig 1. Interaction of EhRho1 with EhGEF. (A)** GST pull down assay was performed from whole cell lysate of HA-EhGEF cell line. HA tag EhGEF was pull down with recombinant wild type GST-EhRho1, GST-EhRho1-Q78L and GST-T34N using Glutathione Sepharose beads and EhGEF was detected using anti HA-tag antibody in western blot. **(B)** Co-immunoprecipitation of EhRho1 and EhGEF from whole cell lysate of *E. histolytica* using Protein-A agarose beads conjugated with anti EhRho1 or anti HA antibodies respectively, as marked on the figure panel. Pre-immune sera were used as control immunoglobulin. **(C)** and **(D)** Lipid overlay assay with HA-tagged proteins expressed in *E. histolytica* trophozoites. The EhGEF corresponds to HA tagged full length wild type protein while ΔEhGEF is HA tagged PH domain only.

activity by catalysing the exchange of GDP for GTP and serve as positive regulators of Rho family proteins by, which results in increased levels of active GTP-bound Rho GTPases in cell [42]. To examine the potential guanine nucleotide exchange activity of EhGEF as well as its specificity for distinct GTPase proteins, a fluorescent GTP based GEF assay was done. HA-tagged EhGEF was immunopurified from the lysate of HA-GEF overexpressing amoebic cell line and recombinant His-tagged EhRho1 and EhRab1a were purified from *E. coli*, were used for fluorescence spectroscopy based GEF assay (**Fig 2A**).The binding of florescent GTP analogue, MANT-GTP (N-Methylanthraniloyl Guanosine 5'-Triphosphate) to protein leads to increased florescence which can reflect protein-nucleotide interaction. It was found that EhGEF was not able to load GTP on EhRab1a but displayed high exchange activity with EhRho1 as shown by increase in florescence (**Fig 2A**). However, EhGEF itself also displayed binding with florescent GTP analogue as seen by increase in florescence, which has been confirmed also later in results. It should be noted that, the rate of increase in florescence for EhGEF alone was slower and lesser than the reaction in which EhGEF was added with EhRho1, which implies that there is a loading of GTP to EhRho1. Also, reaction with EhRho1 and EhRab1A alone showed no GTP binding activity. In order to further confirm the nucleotide exchange, dissociation with MANT-GDP was measured in presence of EhGEF[47]. EhRho1 showed the highest dissociation compared to EhRab1a or HsRhoA in presence of EhGEF. EhGEF exchanged 2-fold faster dissociation of MANT-GDP from EhRho1 compared to HsRhoA and EhRab1a (**Fig 2B**). Results from these experiments suggested that EhGEF specifically interacted with EhRho1 to facilitate GDP exchange with unlabelled GTP. This analysis corroborated the preferential activation of the small GTPase EhRho1 by the EhGEF factor of *E. histolytica*.

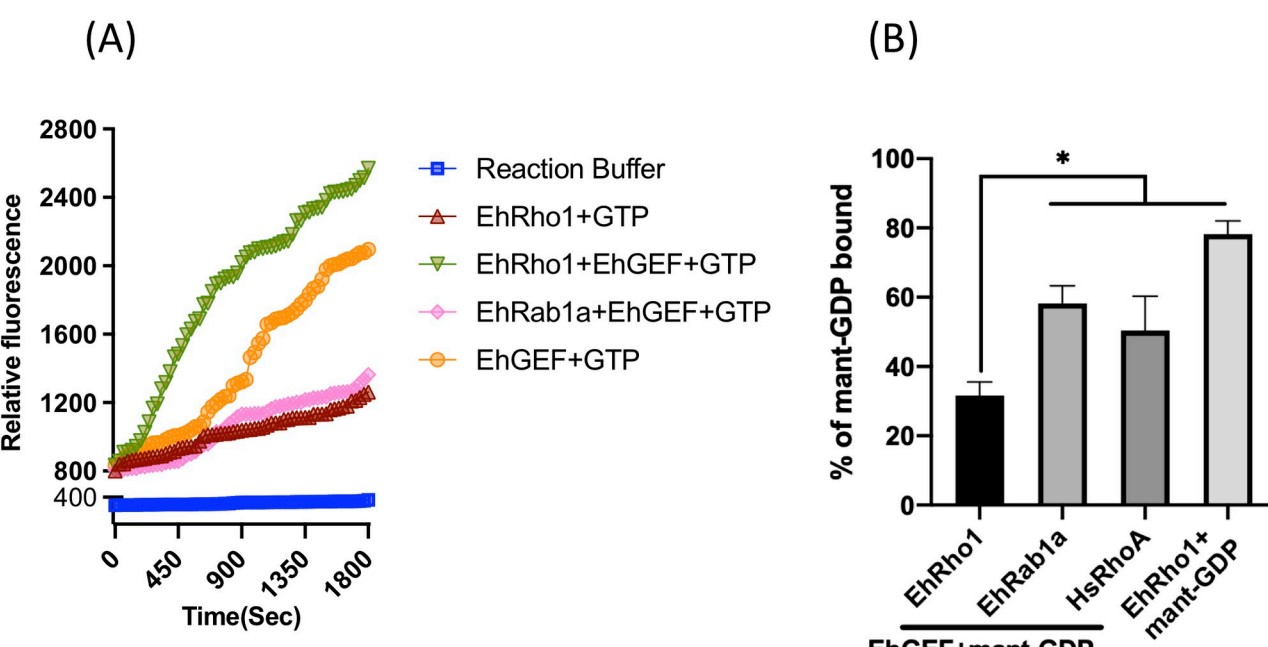

**Fig 2. EhGEF functions as guanine nucleotide exchange factor. (A)** Time-dependent study for association of MANT-GTP from purified recombinant EhRho1 and GST-EhRab1a by HA-EhGEF1. **(B)** Stimulation of MANT-GDP dissociation assays from different small-GTPases by HA-EhGEF using indicated recombinant proteins. Each point represents the mean of triplicate determinations ± SD of the respective means. Statistical differences (*) are shown (t test p < 0.01).

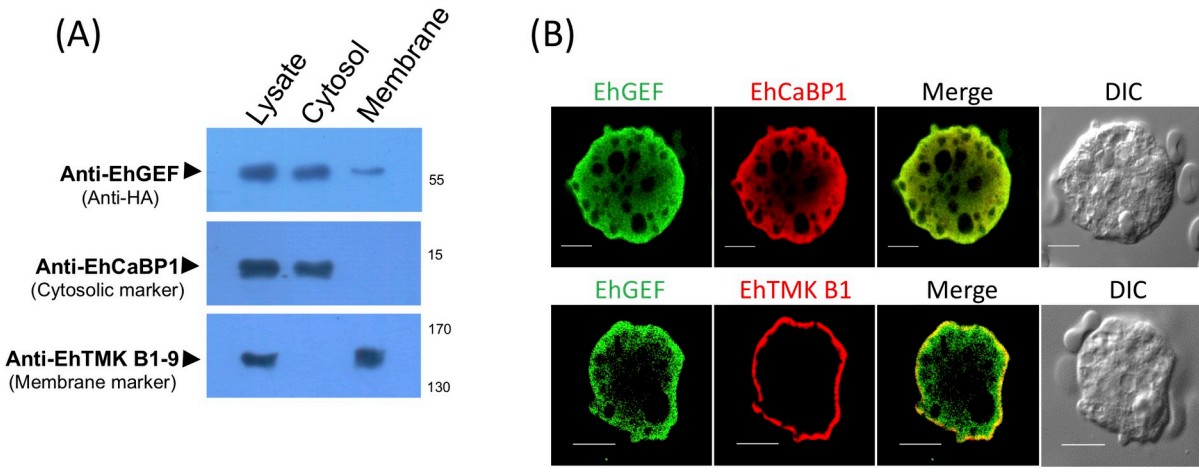

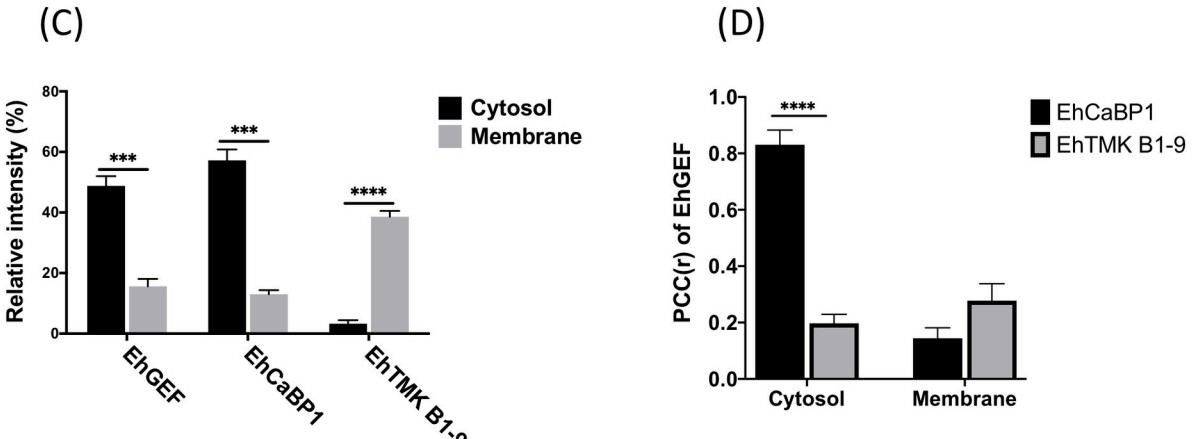

**Fig 3. Cellular localization of EhGEF. (A)** Subcellular fractionation of *E. histolytica* cell lysate was done by using ultra-centrifugation as described in 'experimental methods'. 100μg proteins of each fraction was separated on SDS-PAGE, transfer on PVDF membrane and immunoblotted using anti-HA-tag specific antibody. EhCaBP1 and EhTMKB1-9 specific antibodies were used to determine corresponding proteins as cytosolic and membrane fraction markers respectively in western blot analysis. **(B)** Amoebic cells immunostained for EhGEF, EhCaBP1 (cytosolic marker) or EhTMKB1-9 (membrane marker) using specific antibodies followed by Alexa-488 and Alexa-555 labelled secondary antibodies respectively. **(C)** Quantitative analysis of fluorescent intensity of immunostained cells in panel "B." Five random cells were selected and intensity was taken from multiple sites of membrane and cytosol for EhGFE, EhCaBP1 or EhTMKB1-9. Average relative intensity was calculated by taking the signal from membrane and cytosol as 100% for each marker separately (N = 5). **(D)** Correlation analysis of five cells was carried out by software NIS-Elements AR Analysis 4.00.00. Values of Pearson Correlation coefficient(r) (PCC) have plotted for EhGEF with respect to EhCaBP1 and EhTMKB1-9. Bar represent 10μm, DIC is differential interference contrast. ANOVA test was used for statistical comparisons.*p-value≤0.05, **p-value≤0.005, ***p-value≤0.0005.

## Localization of EhGEF in *E. histolytica* trophozoites

Next, we carried out subcellular localization by fractionation through ultracentrifugation followed by western blotting to investigate localization of EhGEF in amoebic cell lysate. Results from immunoblots of fractionations revealed the presence of EhGEF in both cytosol and membrane fractions (**Figs 3A** and **S1**). To further confirm the result from fractionation experiments, confocal imaging was carried out. Anti-HA antibody was used to visualize the HA tagged EhGEF and specific antibodies were used for EhCaBP1 and EhTMKB1-9 as cytosolic and plasma membrane markers, respectively. It is clear from the images that EhGEF is present

in both plasma membrane and cytosol (**Fig 3B**). In order to further enumerate the localization of EhGEF, a quantitative images analysis was carried out by measuring the pixel density in define ROI. Quantification results showed that intensity of EhGEF is approximately 2.5 fold higher in cytosol than the membrane. The cytosolic expression of EhGEF was comparable to EhCaBP1, however 2 fold lesser extent than EhTMKB1-9 in plasma membrane (**Fig 3C**). We have also estimated the strength of co-localization by formulating Pearson's correlation coefficient (PCC) for fluorescent signals of a pair of stains. We found preferential localization of EhGEF in EhCaBP1 enriched areas (r = 0.898) and as well as in plasma membrane (r = 0.483) with EhTMKB1-9 (**Fig 3D**). Whereas cytosolic marker protein, EhCaBP1 did not show significant colocalization with membrane marker EhTMK B1-9 in cytosol and vice versa also true for TMK B1-9 in case of membrane. The results suggest that EhGEF is likely to be a cytoplasmic protein in inactive state but some proportions binds to plasma membrane through PH domain in normal trophozoites during endocytic processes occurring at basal rate.

## EhGEF is involved in amoebic phagocytosis

EhRho1 has been shown to localize in phagocytic cups by fluorescence microscopy analysis [35] and EhGEF was found in mass spectrometric analysis of immunoprecipitated fraction of EhRho1. In previous reports other amoebic EhGEFs have been reported to participate in phagocytosis [48,49]. Therefore, we were curious to check the involvement of EhGEF in various amoebic phagocytic processes. We used different approaches to test the participation of EhGEF in amoebic phagocytic process. Firstly, immunostaining was carried out to determine the localization of HA tagged EhGEF during human RBC uptake at different time points, using specific antibody against the HA-tag. It was clearly shown that EhGEF start accumulating in phagocytic cups at 3 minutes of RBC addition to cells and remain until the closure of phagocytic cup but disappears soon after scission of phagosome from membrane at 7 minutes. The enrichment was also observed in the leading tips of the pseudopods. Actin was also observed to be enriched at the site of phagocytic cups along with EhGEF. Complete superimposition of EhGEF with both EhActin suggested that both proteins are colocalized at the phagocytic cups during progression of cup (**Fig 4A** and **4B**). Results from this experiment suggests the possible involvement of EhGEF in amoebic phagocytic process.

In order to understand the role of EhGEF in the context of some of the other molecules that have been identified as part of the phagocytosis pathway in *E. histolytica* (EhCaBP1, EhC2PK, EhCaBP3), pairwise staining was carried out and extent of co-localisation during phagocytosis was quantified using PCC (**S2A–S2F Fig**) [50–53]. All marker proteins, namely, EhCaBP1, EhCaBP3, EhC2PK[53] and EhActin were enriched in phagocytic cups as previously reported [54]and were found to colocalise with EhGEF (**S2A and S2B Fig**). However, both EhCaBP1 and EhC2PK disappeared from the phagocytic cups immediately following the closure in manner similar to EhGEF while EhCaBP3 was the only molecule present in late endosome. EhGEF was not observed to colocalized with EhCaBP3 in late phagosomes but, more likely disappear just immediately after scission takes place (**S2C Fig**). Comparison of immunostained images of EhGEF and EhTMKB1–9, a general plasma membrane marker, clearly supports specific enrichment of EhGEF in phagocytic cups, which can be seen in case of EhCaBP1, EhC2PK, and EhCaBP3.

To further investigate the real time cellular distribution of EhGEF during phagocytosis a N-terminal GFP tagged EhGEF was expressed in amoebic cells. The expression of GFP-EhGEF was confirmed by immunoblot using anti-GFP specific antibody as shown in **S3A Fig**. To evaluate any potential mis-localization of GFP-EhGEF due to GFP tag, these cells were immunostained using anti-GFP antibody. GFP-GEF showed the similar distribution and enrichment at the phagocytic cups as HA-EhGEF whereas vector control cells which expressing GFP only did

(A)

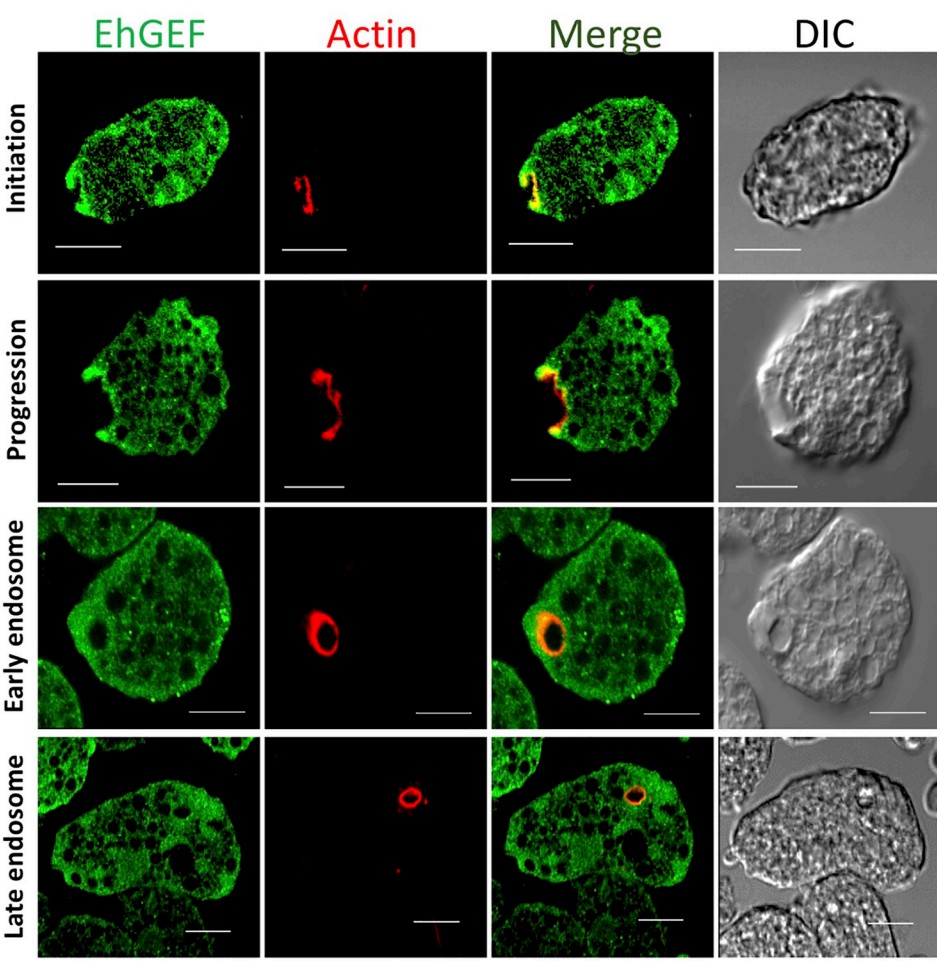

(B)

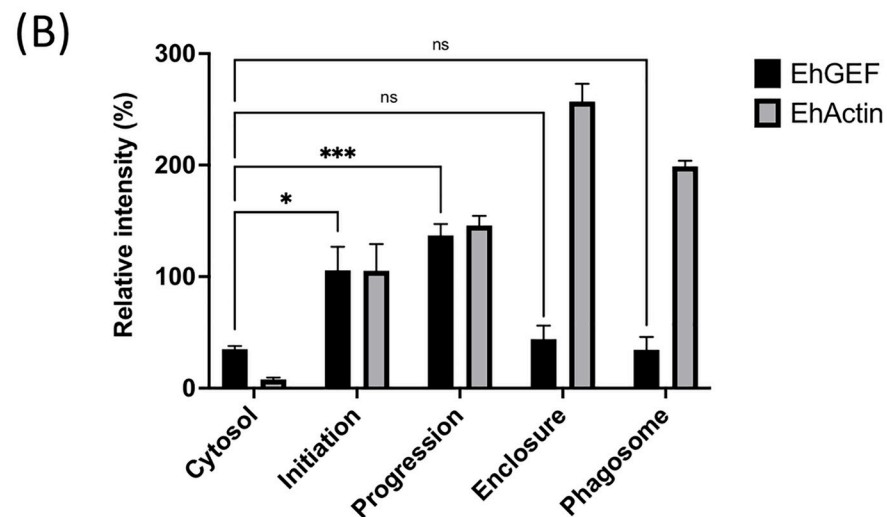

**Fig 4. EhRho1 is involved in Phagocytosis. (A)** Imaging of EhGEF and actin during erythrophagocytosis assay. *E. histolytica* cells (after 48h) were harvested and incubated with RBC for indicated time intervals at 37˚C and subsequently fixed for further processing. EhGEF protein were immunostained with HA tag specific antibody followed by Alexa-488 conjugated secondary antibody (green). F-actin was stained with TRITC conjugated phalloidin (red). Slides were viewed using Nikon confocal microscope. Arrowheads indicate phagocytic cups and asterisks marks enrichment of actin in phagosome. **(B)** Quantitative analysis of relative pixel intensity of fluorescent signals from EhGEF and EhActin were calculate from indicated sites of actively phagocytosing amoebic cells. Relative intensities were calculated by NIS-Elements AR 3.0 and plotted intensity as 100% for each marker separately. This experiment was carried out by selecting randomly twenty cells, (N = 20, bar represent standard error).Scale bar, 10 μm; DIC differential interference contrast.

not show any enrichment in phagocytic cups (**S3B Fig**). Using these GFP-EhGEF expressing trophozoites a time-laps imaging was performed in modified erythrophagocytosis assay [35,48]. Here, RBCs were stained with red cell tracker dye (Red CMTPX Dye) as manufacturer protocol [55]. The data are shown in the form of montages of complete cycle of phagocytosis at different intervals of time-lapsed movie (**Fig 5A and S1 Movie**). GFP-EhGEF was observed to be rapidly enriched in the plasma membrane of trophozoite in contact with RBC and remain there till closure of phagocytic cups. Florescence intensity of GFP-EhGEF along the arrow drawn across the phagocytic cup (inset figure of **Fig 5B**) at different time intervals showed enrichment of GFP-EhGEF to the plasma membrane in contact with RBC, while the rest of the plasma membrane and cytoplasm showed low levels of GFP-EhGEF (**Fig 5B**). Following closure of phagocytic cup, quantitative analysis of the GFP-EhGEF along the drawn arrow in inset of **Fig 5C**, showed drastic reduction in florescence intensity in the membrane spanning phagocytic cup (**Fig 5C**).

EhGEF was also found to be involved in phagocytosis of other cell types, such as dead mammalian cells, which we used heat killed CHO cells as an example. The parasite specifically engulfs dead or apoptotic cells via phagocytosis only and not trogocytosis [56]. This was visualized by observing enrichment of EhGEF during phagocytosis of dead Chinese hamster ovary cells (CHO) labelled with Cell Tracker Red dye and by time lapse imaging (**Fig 5D and S2 Movie**). EhGEF was observed at the phagocytic cups from the start of the phagocytosis till closure of phagocytic cups. The florescence intensity along the line across phagocytic cups confirmed enrichment of GFP-EhGEF to the membrane spanning phagocytic cup and in contact with dead CHO cell (**Fig 5E**), while intensity of EhGEF dropped significantly once the phagocytic cup closed (**Fig 5F**). Interestingly, here also we observed that EhGEF was present just after the membrane fusion event but not when phagosome got separated from the membrane. In phagocytosis of RBC and dead CHO cells, the enrichment followed a zipper like pattern as phagocytic cup progressed to engulf the ligand and soon after scission of the phagosome, the molecule disappeared (which is evident from the montages).

As EhGEF has been shown to interact with EhRho1, the localisation of both the proteins was assessed by measuring co-localization in images of immunostained phagocyting amoebic cells expressing HA-GEF. EhGEF was found to coexist with EhRho1 at the phagocytic cups as well as at the progression of phagocytic cups (**Fig 6A and 6B**). Interestingly, we observed that the localization of EhGEF became different from EhRho1 just after the membrane fusion event, where EhRho1 was present in phagosome, but HA-EhGEF left the site. Quantitative analysis (PCC) of the images further support that both molecules co-localize throughout the phagocytic events except on enclosed phagosome (**Fig 6C**).

## EhGEF regulates the amoebic phagocytosis

For functional analysis of EhGEF, the overexpression and downregulation of gene expression was carried out by expressing sense and anti-sense mRNA [57,58].The expression of EhGEF

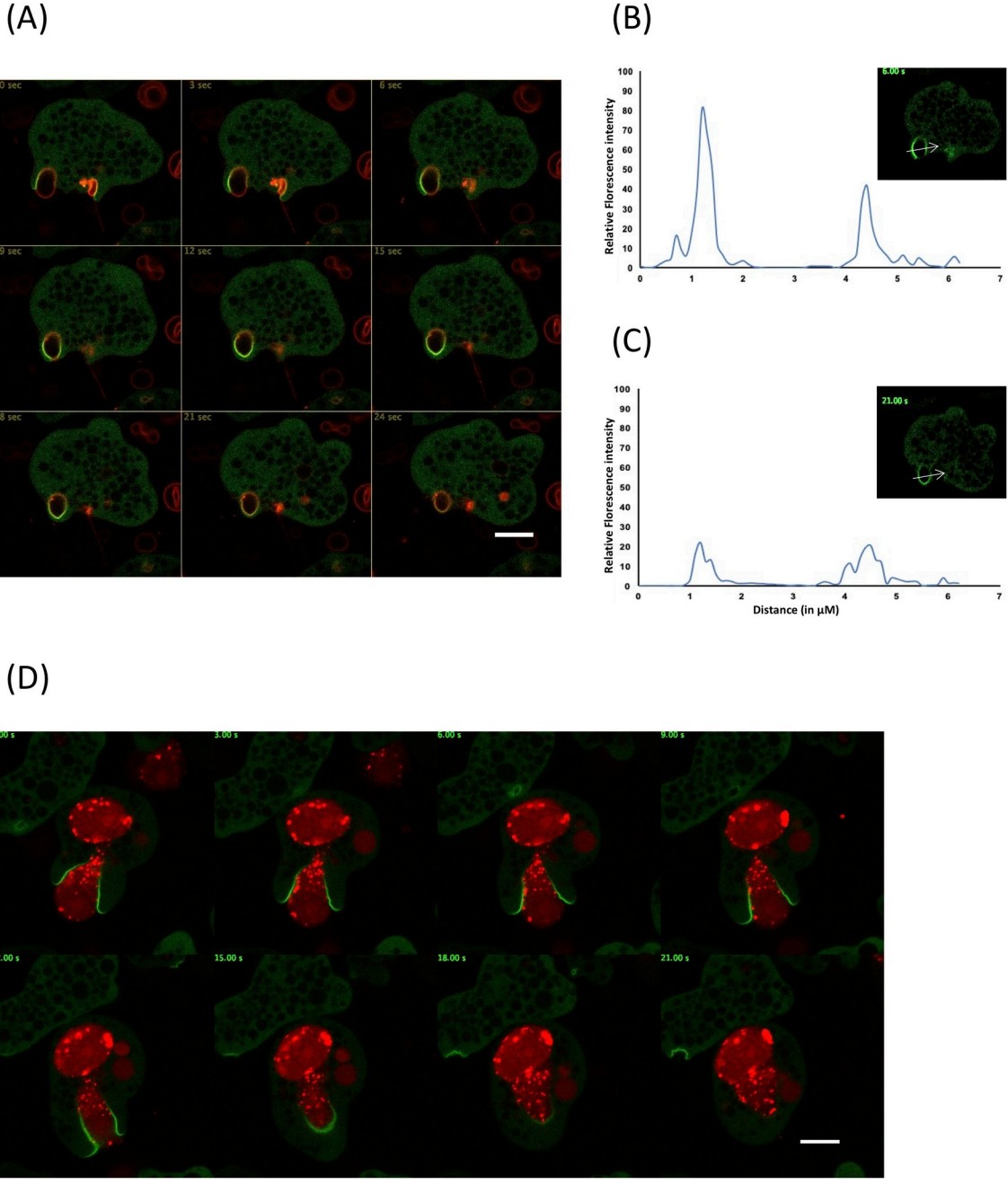

**Fig 5. Live cell imaging of GFP-EhGEF expressing *E. histolytica* cells during phagocytosis. (A)** The montage of trophozoite expressing GFP-EhGEF engulfing labelled human RBC. **(B and C)** Analysis of intensity of GFP-EhGEF along the arrow-line marked across the phagocytic cup in inset figure **(B)** and newly formed phagosome **(C)**. **(D)** Time laps montage of *E. histolytica* cell expressing GFP-EhGEF during phagocytosis of dead CHO cells respectively. The montage shows a time series of GFP-EhGEF expressing cells undergoing phagocytosis where phagocytic cups are marked by arrowhead and just closed phagosomes by star. **(E and F)** Graph shows the intensity of GFP-EhGEF along the line shown in inset figure and reveals enrichment specifically in plasma membrane spanning phagocytic cup **(E)** while the intensity decreases in newly formed phagosome **(F)**. Scale bar, 10 μm.

was increased by transfecting the plasmid encoding the gene in sense orientation in tetracycline inducible pEhHyg-TetR-O-CAT (TOC) vector. Induction with tetracycline led to an increase in protein level by about 40% in cells expressing the gene in sense orientation with

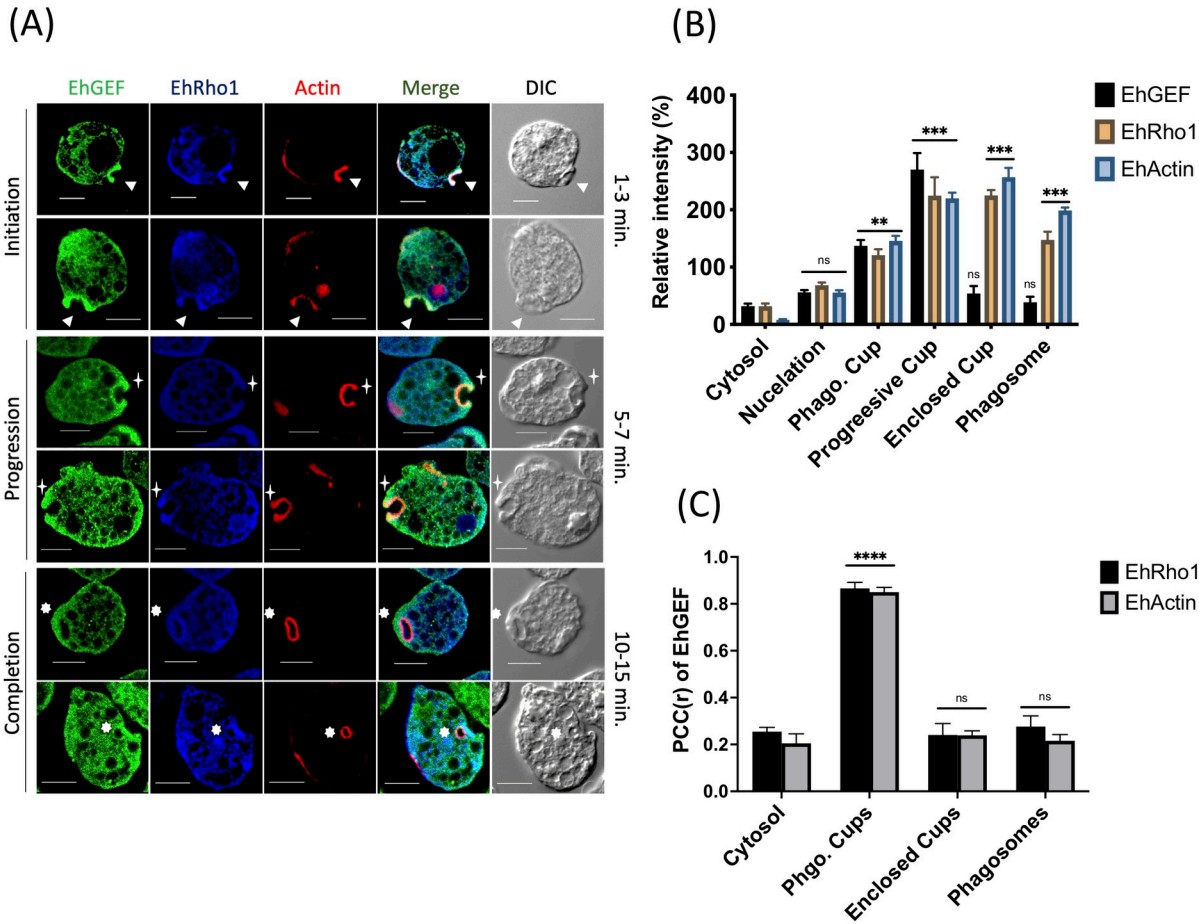

**Fig 6. EhGEF recruited to phagocytosis along with EhRho1. (A)** Florescence images of EhGEF (green) with F-actin (red) and EhRho1 (blue) during erythrophagocytosis. Cells were immunostained with anti-HA tag and EhRho1-specific antibody followed by Alexa-488 (green) and Alexa-405 (blue) conjugated secondary antibody, respectively. F-actin was stained with TRITC conjugated phalloidin (red). Arrowheads indicate phagocytic cups and asterisks mark enrichment of actin in phagosomes. **(B)** Quantitative analysis of fluorescent signals of indicated proteins at the specific site was done as described in materials and methods, which shows loss of signal intensity of EhGEF in newly formed phagosome in comparison to phagocytic cup. **(C)** Pearson correlation coefficient(r) analysis of indicated proteins at the specific site was performed. Scale bar indicates 10 μm, DIC = differential interference contrast.

HA-tag at N terminus, as determined by densitometric scanning of the immunoblots. Here EhCoactosin have been used as a loading control (**Fig 7A**). The HA-tag was added to identify the ectopically expressed EhGEF upon induction with tetracycline. Further, the first 500bp of EhGEF was cloned in anti-sense direction in tetracycline inducible pEhHyg-TetR-O-CAT (TOC) vector and stable cell lines for conditional knockdown of EhGEF was generated in *E. histolytica* (**S4A Fig**). Transfected parasites were selected and grown at 10 μg/ml of hygromycin. The level of down regulation was determined by real time PCR at different tetracycline concentrations as shown in **Fig 7B**. The expression of EhGEF was reduced by more than 50% in the presence of 30 μg/ml tetracycline (Tet(+)).

Transfected *E. histolytica* cells carrying the sense and antisense (AS) constructs were then checked for erythrophagocytosis using a spectrophotometric assay. All comparisons were made against cells carrying either the vector alone, or with the gene construct in the absence of tetracycline. There was a 60% reduction of erythrophagocytosis in cells expressing EhGEF antisense RNA (that is, in the presence of tetracycline) as compared with cells carrying only

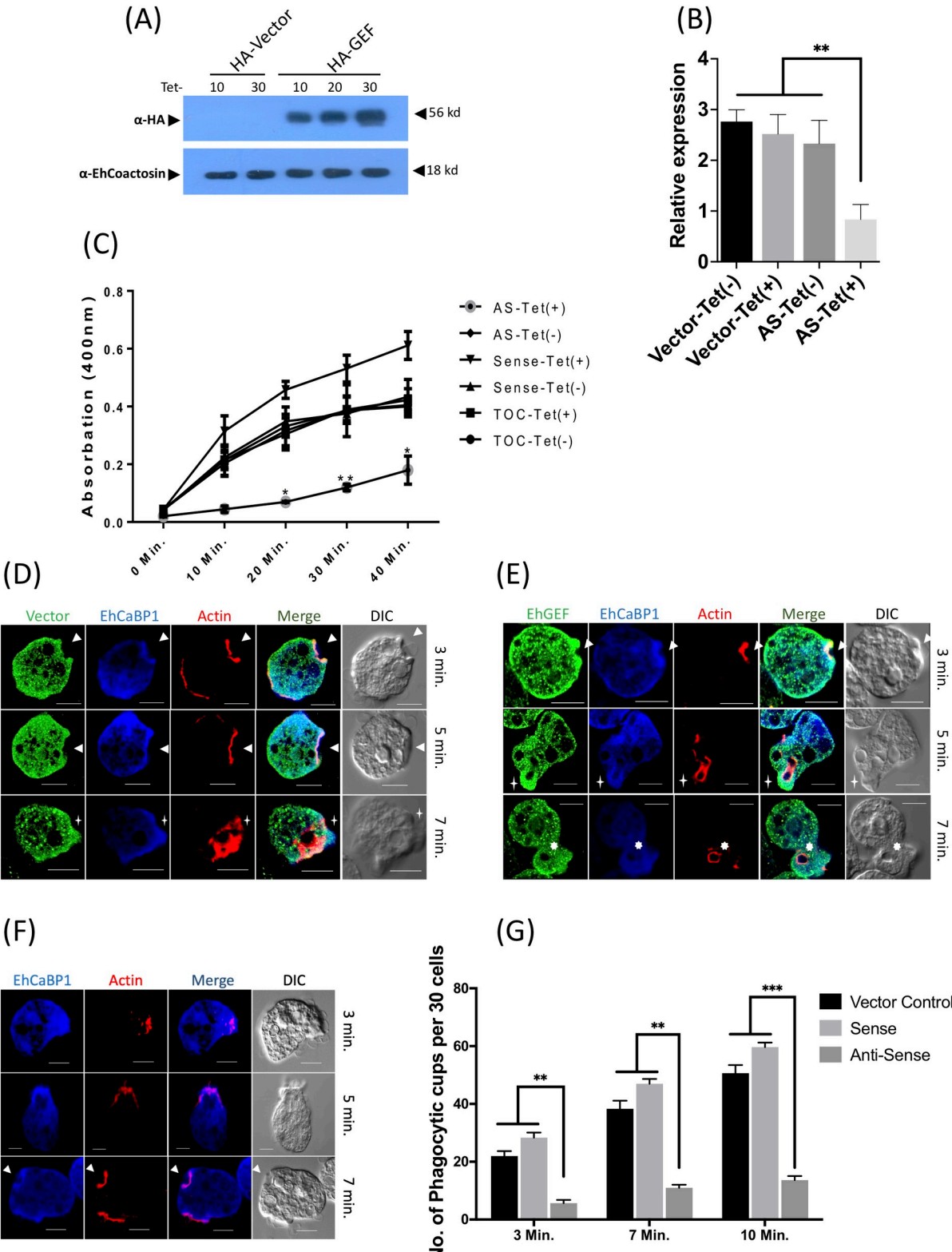

**Fig 7. EhGEF regulates the phagocytosis in *E. histolytica* cells. (A)** HA tagged EhGEF expression was determined in presence or absence of tetracycline by immunoblotting using anti HA-tag specific antibody. EhCoactosin was used as loading control. **(B)** EhGEF expression was quantified in anti-sense cell line of EhGEF by real-time PCR. **(C)** Spectrophotometric assay for erythrophagocytosis. RBC uptake assay was performed in cells expressing sense and anti-sense EhGEF constructs in presence and absence of tetracycline. RBC were incubated with

indicated cell lines for different time points and the amount of RBC uptake was determined spectrophotometrically using RBC solubilisation assay as described in 'experimental methods'. The experiments were carried out three times independently in triplicates. **(D, E & F)** Amoebic cells carrying vector alone (D), sense (E), and anti-sense (F) constructs in tetracycline inducible vector, were grown for 48h in presence of 30μg tetracycline and incubated with RBC for indicated times. EhGEF and EhCaBP1 were immunostained with anti HA-tag and anti EhCaBP1 antibodies followed by secondary antibodies conjugated with Alexa-488 and Alexa-405 respectively. EhActin was visualized by TRITC-phalloidin staining. Solid arrow showed the phagocytic cups. **(G)** Quantitative analysis of phagocytic cups. Randomly 30 cells were selected in three sets for each experiment and number of phagocytic cups present in selected cells were counted for indicated cell lines.Scale bar indicated 10μm, DIC is differential interference contrast. ANOVA test was used for statistical comparisons.*p-value≤0.05, **p-value≤0.005, ***p-value≤0.0005.

the vector in the presence of tetracycline, and cells carrying EhGEF antisense construct in the absence of tetracycline. Overexpression of EhGEF, by tetracycline induction of the transfectants carrying a sense construct displayed an increase (30%) in erythrophagocytosis as compared to cells without tetracycline or vector containing cells in the presence of tetracycline (**Fig 7C**). The observed reduction in phagocytosis on down regulation of EhGEF expression may be due to either a reduction in initiation, progression or scission of phagosome formation. In order to identify the steps affected, cells expressing vector control or EhGEF sense and anti-sense RNA were incubated with RBCs for indicated time and analysed by immunostaining. Cells expressing vector only showed phagocytosis with normal rate (**Fig 7D**). In comparison, many phagocytic cups were visible in cells over expressing EhGEF by 3 min (**Fig 7E**). However, in antisense EhGEF RNA expressing cells phagocytic cups were visible only after 5 min of incubation with RBC. We scarcely observed phagosomes in these amebic cells after 7 minutes of incubation (**Fig 7F**). We carried out quantitative analysis of the images by observing 30 cells (in triplicates) and enumerated number of phagocytic cups and phagosomes in these cells. By 5 min, cups and phagosomes were found to be only 13% and 6% of the control cells in anti-sense cells respectively (**Fig 7G**). The results clearly showed that compared with vector control cells with tetracycline, the rates of both cup and phagosome formation were significantly reduced in cells downregulated for EhGEF expression. On the other hand, in cells over expressing EhGEF, phagocytic cups and phagosomes increased by 65% and 30% respectively. From this data we can conclude that EhGEF is involved in phagocytosis by *E. histolytica*. Similar results were obtained when cell proliferation was measured. A significant reduction in proliferation was observed in cells expressing anti-sense mRNA of EhGEF in comparison to vector control and/or construct without tetracycline induction (**S4B Fig**), consistent with our previous finding with EhRho1[35]. This also reflects, impaired endocytic processes hampered the growth of trophozoites due to nutritional deficiency in axenic culture conditions.

## EhGEF regulates amoebic phagocytosis by modulating EhRho1 activity

The interaction between EhRho1 and EhGEF was confirmed by co-immunoprecipitation of EhGEF using EhRho1 specific antibody in total *E. histolytica* cell lysate expressing HA-tagged EhGEF followed by western blot analysis using anti HA antibody and EhRho1 specific antibodies. To elucidate the role of EhGEF in EhRho1 in activation during phagocytosis, the levels of active EhRho1 were estimated in sense and anti-sense EhGEF expressing trophozoites. For this purpose, Glutathione S-transferase (GST) fused-Rho-binding domain (RBD) of Rhotekin, known to interact specifically with active form of HsRhoA was used as positive control [25,30,59,60]. Binding of Rhotekin sequesters Rho proteins in GTP bound form and inhibits GAP-stimulated or intrinsic GTPase activity of Rho[61]. The specificity of binding of GST-Rhotekin RBD with GTP-EhRho1 has been shown before[30]. Cell lysates from these above indicated cell lines were subjected to GST-Rhotekin pull down and subsequent analysis by western blotting (**S5A Fig**). The amount of active EhRho1 decreased in cells expressing

anti-sense RNA of EhGEF. However, in EhGEF overexpressing cells, a noticeable increase in the amount of GTP-bound EhRho1 was observed through specific antibodies. There was no effect on the total level of EhRho1. These results indicated that EhGEF regulates the activation of EhRho1 and might be playing a role in regulation of phagocytosis.

Following activation of EhRho1 by EhGEF, we focussed on involvement of EhGEF in recruitment of EhRho1 and its downstream partners, EhFormin1 and EhProfilin1 to the phagocytic cups. As PH domain of EhGEF binds PtdIns(3,4,5)P3 which is well known to be generated during endocytic processes, EhGEF might also play role in recruiting EhRho1 to phagocytic site. In order to demonstrate this, we visualized the sub-cellular localization of EhRho1, EhFormin1 and EhProfilin1 in RBC phagocytosing cells transfected with antisense constructs of EhGEF in the presence and absence of tetracycline. The analysis of immunos-tained images show that EhRho1 was not enriched at the site of RBC attachment in anti-sense EhGEF cell line grown in presence of tetracycline when incubated with human RBC (**Fig 8A**). However, in cells transfected with vector alone or HA-GEF, actin was seen predominantly in phagocytic cups, while EhRho1 and EhGEF1 were visible in both cytosol and phagocytic cups. Quantitative analysis indicated 60% reduction in the level of EhRho1 signal at RBC attachment sites in EhGEF antisense cells in presence of tetracycline as compared to level of EhRho1 signal at the phagocytic cup in vector alone or HA-EhGEF expressing cells in presence of tetracycline. As reported earlier, EhRho1 dominant negative mutant (T34N) which is not capable of bind-ing GTP, shows depletion effect from phagocytic sites that is comparable with mislocalization pattern of EhRho1 in anti-sense EhGEF cells. Similar results were obtained with EhFormin1 and EhProfilin1 when EhGEF levels were down regulated in antisense cell line in presence of tetracycline (**S5B Fig**).The effect was specific between EhGEF and EhRho1, as in EhCaBP1 down-regulated cells there was no significant difference in the recruitment of EhRho1 (ratio of cups to cytosol, 2.5) (**Fig 8B**). In order to rule out any possibility of down-regulation of EhRho1 protein expression in EhGEF antisense cells, we investigated the levels of EhRho1 in EhGEF antisense protein expressing cell lines by immunoblotting. We did not observe any change in the level of EhRho1 in these cells (**S5C Fig**). These results clearly suggest that EhGEF is involved in phagocytosis and recruits EhRho1 to phagocytic cups.

## EhGEF is also involved in pathogenesis

There are many steps involved in cytolysis and subsequent phagocytosis of target cells by *E. histolytica*. These steps are, recognition, adhesion, cytolysis followed by phagocytosis of dead target cells. Invasive infection of *E. histolytica* starts with adhesion to colonic epithelial substra-tum cells. The role of EhGEF on *E. histolytica* target cell adhesion was investigated by using CHO cell adhesion assay [62]. There was a 30% reduction in cells expressing reduced amount of (anti-sense) EhGEF. However, over-expressing HA-EhGEF cells displayed enhancement (45%) in adhesive property as compared to cells carrying only vector (**Fig 9A**). Similar results were also obtained when cytopathic property was determined. There was about 40% decrease in ability to destroy CHO monolayer by cells expressing anti-sense EhGEF as compared to vec-tor alone. Further a significant 20% increase in cytopathic activity was observed in cells expressing HA-GEF protein in trophozoite cells (**Fig 9B**). These results suggest that EhGEF is actively involved in *E. histolytica* pathogenesis and may play an important role during host tis-sue invasion.

The involvement of EhGEF in destruction of live cells via trogocytosis was further demon-strated by real time imaging. The GFP-EhGEF distinctly localised to the tips of the membrane involved in pinching of the live host cells labelled with red cell tracker dye (**Fig 9C** and **S3 Movie**). The GFP-EhGEF localised to the tips of the tong like structures formed by *E.*

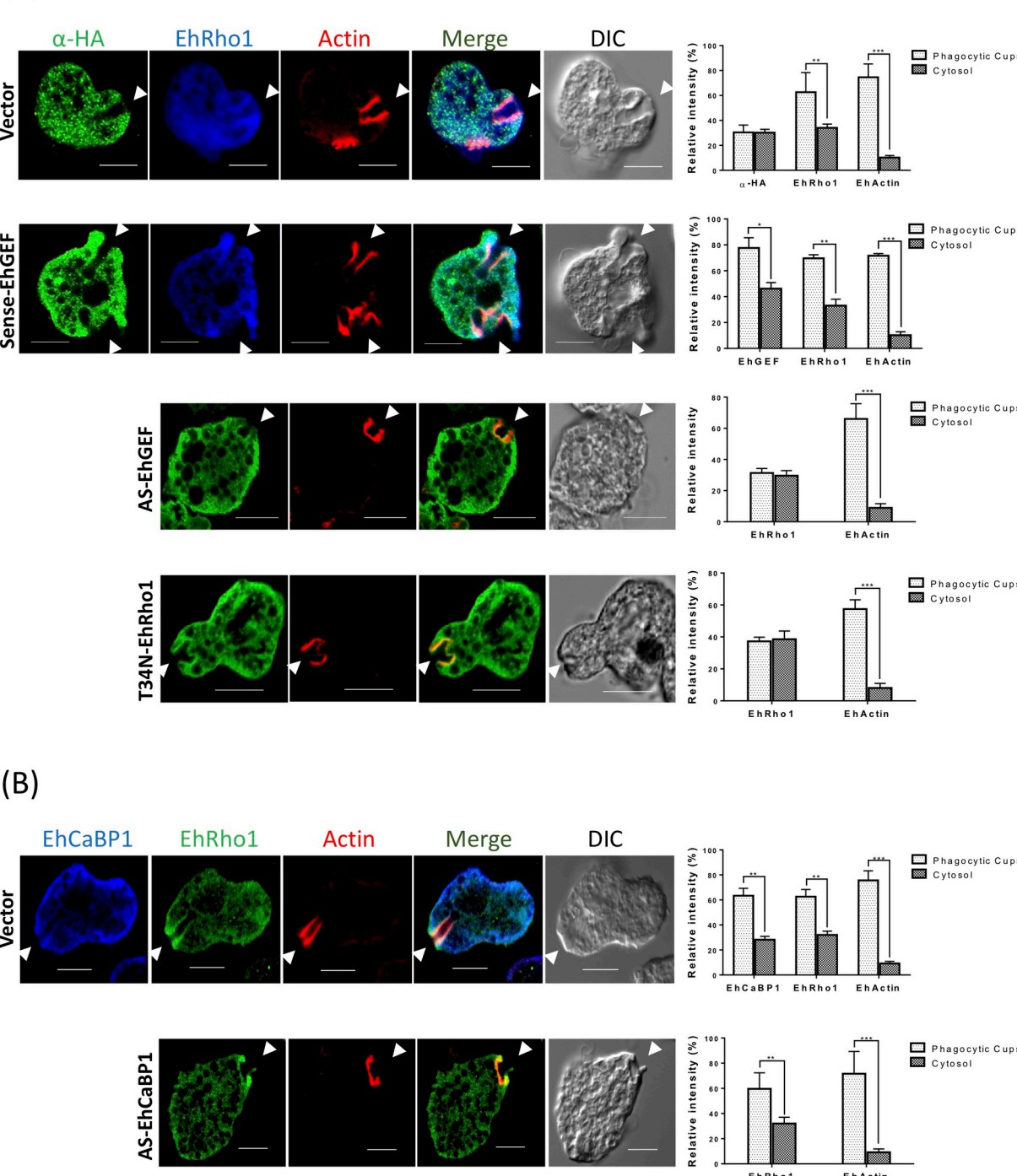

**Fig 8. EhRho1 recruited through EhGEF. (A)** *E. histolytica* cells expressing indicated constructs were grown in presence of 30μg/ml tetracycline and incubated with RBC for 5 min at 37˚C. Cells were fixed and immunostained with indicated protein-specific antibodies. Alexa-488 (green) and pacific blue-410 (blue) were used for immunostaining of proteins and F-actin was stained with TRITC phalloidin (red). Arrowheads indicated phagocytic cups with enrichment of indicated proteins and asterisk marks the just closed cup before scission and star represent newly form phagosomes. Transfectants downregulated for EhGEF expression (EhGEF-AS) and overexpressing T34N mutant of EhRho1, both showed dominant negative phenotype. Graphs represent the quantitative analysis of fluorescent signal from cytosol and phagocytic cups of immunostained images of EhGEF, EhRho1 and EhActin containing cells. Five random regions were selected in five cells for analysis of fluorescent signal from cytosol and phagocytic cups of indicated cells and average fluorescent intensity was calculated for each region (N = 5, bar represent standard error). Relative intensity was calculated by assuming intensity as 100% for each marker separately. **(B)**

Immunofluorescence images of *E. histolytica* cells containing indicated constructs during erythrophagocytosis. Cells were stained with anti-EhRho1 or anti-CaBP1 antibodies followed by Alexa-405 or Alexa-488 secondary antibodies. Actin was stain with TRITC-phalloidin. The recruitment of EhRho1 is independent of EhCaBP1 mediated signalling pathway. ANOVA test was used for statistical comparisons.*p-value≤0.05, **p-value≤0.005, ***p-value≤0.0005.Bar represent 10μm, DIC is differential interference contrast.

*histolytica* for biting the live host cells. Further, the florescence intensity across the tunnels formed during trogocytosis shows enrichment in the plasma membrane in contact with live CHO cell (**Fig 9D**) while no significant GFP-EhGEF could be observed in the endosomes (can be called trogosome) formed after pinching (**Fig 9E**). This finding clearly shows that it is involved in destruction of live host cells via trogocytosis as well.

## Structure-function relation of EhGEF

**Molecular modelling propounds a PT-barrel structure for EhGEF.** EhGEF does not share sequence homology with other members of the GEF family, and low sequence identity (<20%) was found with proteins having experimentally determined structure. The 3D structure of EhGEF was therefore modelled using various structure prediction tools including Robetta (Rosetta comparative and *ab initio* methods), TrRosseta (neural-network based structure prediction), I-TASSER (threading-based structure prediction) and SWISS MODEL. The structures obtained from all the modelling servers resembled a PT-barrel (first discovered in prenyltransferase) (**Fig 10A**) except from Rosetta *ab initio* which resulted in a model with a distinct fold. PT-barrel is a ten-stranded β-barrel found in prenyltransferases with ααββ structural repeat[63]. The PT-barrel resembles the TIM barrel that the β-sheets are surrounded by the solvent exposed $\alpha$-helices with different secondary structural connectivity. Another major difference is, all β-sheets are antiparallel that one side they are connected with β-hairpin and other side with two consecutive $\alpha$-helices and loops. The barrels create a hole inside with hydrophilic environment where substrate can bind, and the loops that are connecting the β-sheets also participate in the ligand binding. Validation of the models was done based on multiple criteria like stereochemical deviation in protein geometry (MolProbality), z-score provided by ProSA, QMEAN4 score, Verify3D and quality factors provided by ERRAT. Best validation metrics were obtained for model generated by Robetta webserver through Rosetta comparative modelling, therefore it was selected for further docking and simulation studies. GTP binding to EhGEF was modelled through molecular docking of GTP in the cavity predicted by CASTp (**Fig 10B**). Docking results shown that the binding of GTP and modelled EhGEF is mediated by hydrogen bond interactions of GTP with Ser60, Arg62, Ser110, Asn113 and Thr165 (**Fig 10C**) with a predicted binding energy of -9.4 kcal/mol. Molecular dynamics simulations of the modelled EhGEF and the docked complex with GTP allowed assessment of the stability under aqueous conditions and refinement of the docked complex. The equilibrated RMSD profile shows no significant conformational changes in the modelled protein and also the protein-GTP complex is fairly stable under aqueous conditions (**S6A Fig**). The radius of gyration showed that the protein-GTP complex is compact throughout the simulations where EhGEF has maintained its structural integrity in complex with GTP (**S6B Fig**).

**EhGEF belong to α/β class of protein that binds to GTP.** In order to probe the secondary structure of the protein and also confirm the predicted folds experimentally, CD spectroscopy was performed with recombinant protein lacking PH domain (EhGEFΔPH). The percentage secondary structure was estimated using DichroWeb server. The CD spectra propounds that the EhGEF belongs to α/β class of protein. The secondary structure estimation from CD spectra shows close agreement with the secondary structure estimated from protein sequence information and also the modelled structure (**S7 Fig**).

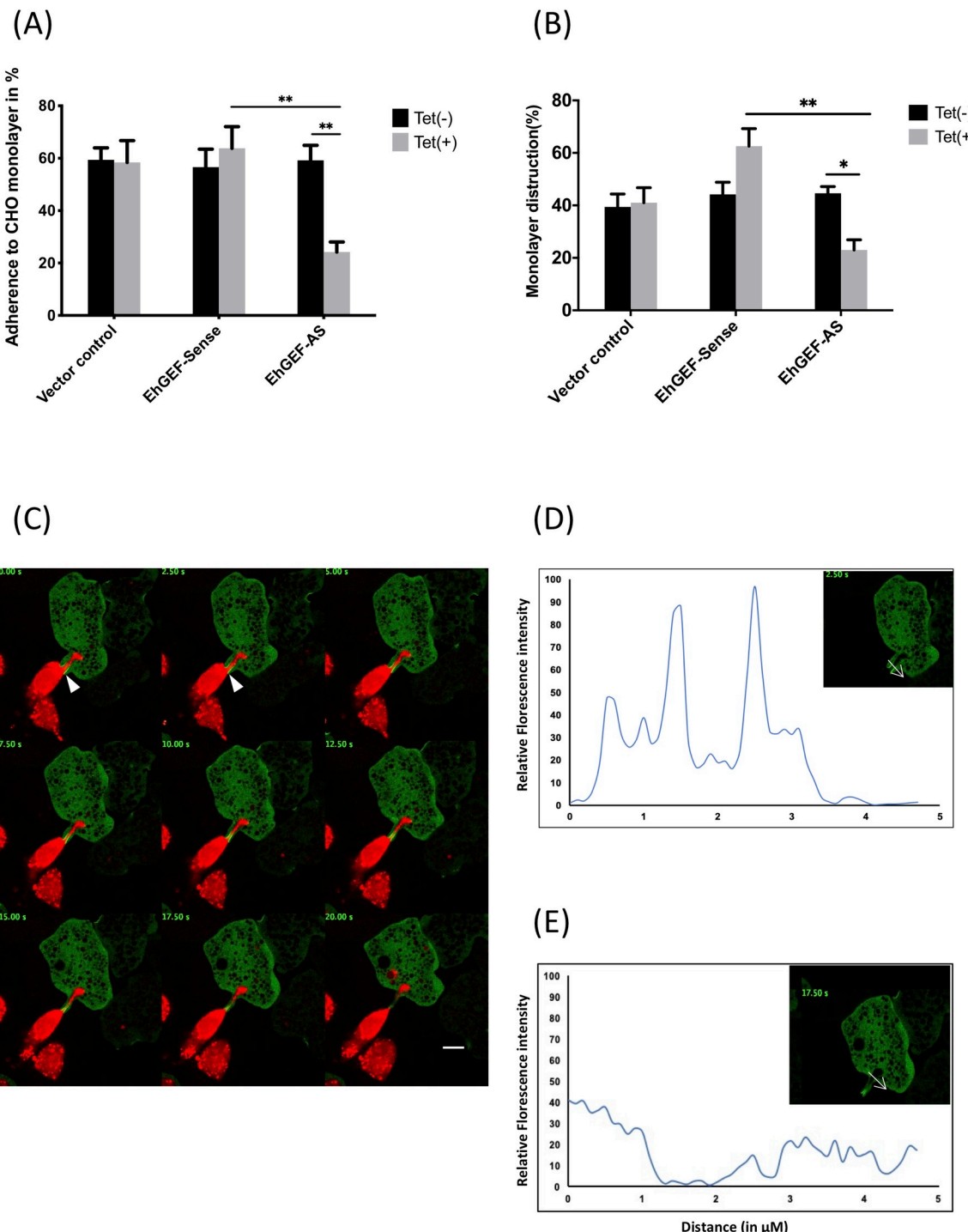

**Fig 9. Involvement of EhGEF in host cell destruction. (A)** Adhesion of EhGEF ovrexpressing and down regulated transfectants (2x10⁵) to monolayers of CHO cells after 30 min of interaction in presence and absence of tetracycline. The trophozoites that remain attached to the monolayer after gentle washing were counted and plotted. **(B)** Destruction of CHO monolayer by indicated EhGEF transfectants. Amoebic and CHO cells were used in 1:1. The killing of CHO cells was determined by counting after methylene blue staining. Results of three independent experiments are shown here. ANOVA test was used for statistical comparisons. *p-value≤0.05, **p-value≤0.005, ***p-value≤0.0005. **(C)** The time-series montage of amoebic trophozoite expressing GFP-EhGEF ingesting live labelled CHO cell (marked by arrow). **(D and E)** The relative florescence intensity profile showing the recruitment of GFP-EhGEF to the plasma membrane in contact with live CHO cell (D), also the enrichment of GFP-EhGEF is lost by the time biting process ceases (E).

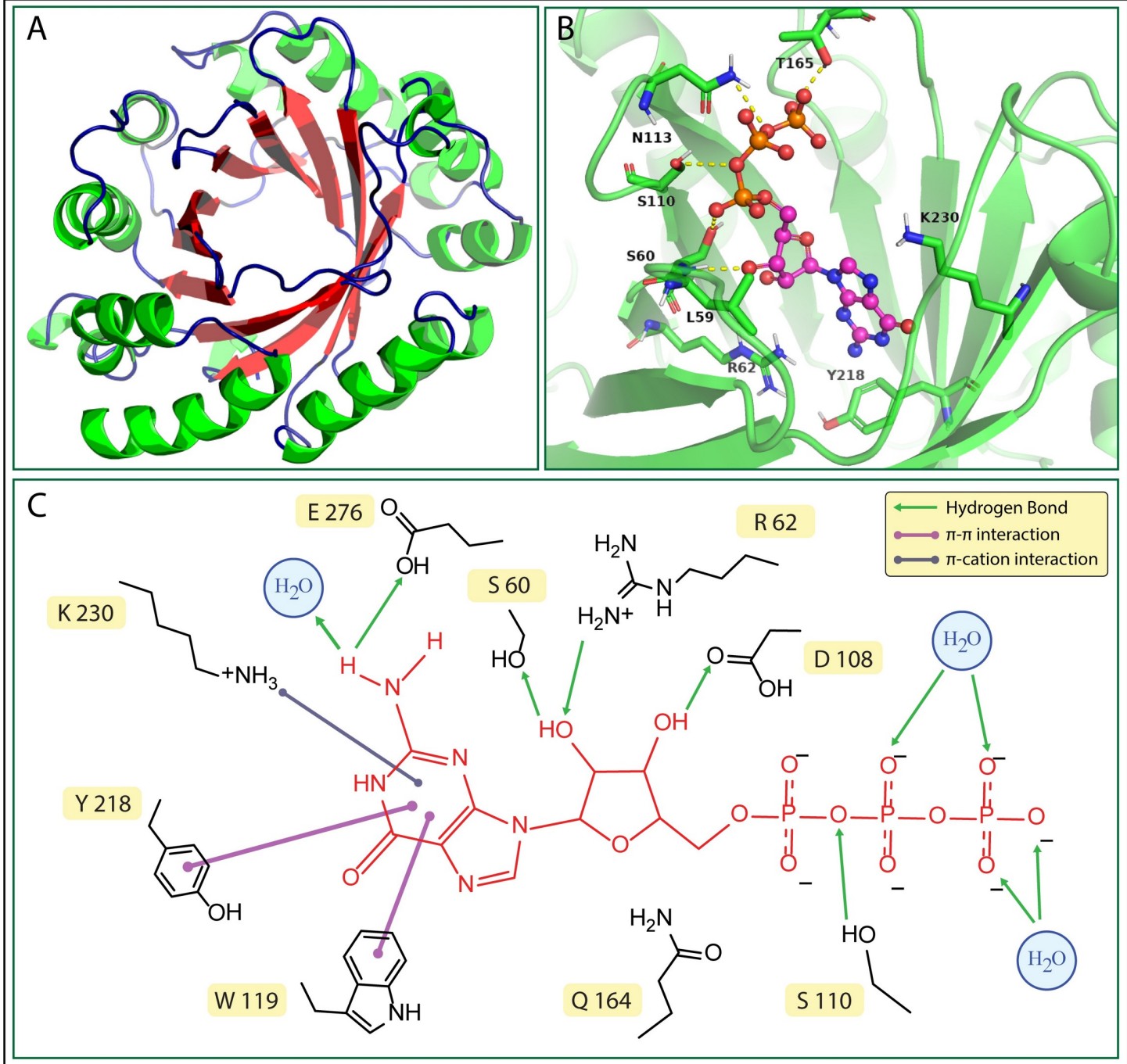

**Fig 10. Molecular modelling and docking. (A)** EhGEF 3D structures model from Rosetta comparative modelling. **(B)** Modelled complex of GTP (ball & stick, magenta) with EhGEF (cartoon, green) and selected residues interacting with GTP are shown as sticks in respective colour. **(C)** Substantial interactions observed during MD simulation of modelled complex of GTP with EhGEF.

As observed in previous results, **Fig 2A**, EhGEF showed GTP binding activity and molecular docking prediction also agrees to the observation. The modelled EhGEFΔPH and GTP complex shows proximity of Trp106 and Trp119 with GTP. Binding of GTP was therefore studied using tryptophan quenching assay. The saturation curve obtained from label free

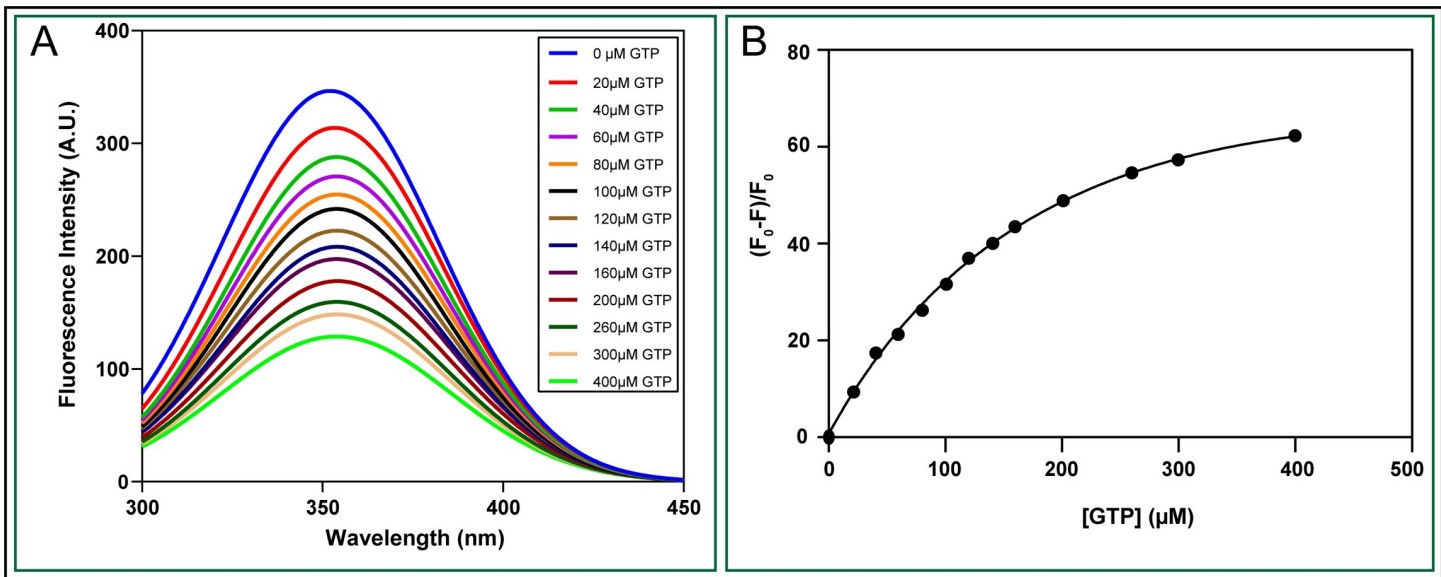

**Fig 11. Determination of the dissociation constant ($K_D$) for GTP with EhGEF by fluorescence spectroscopy. (A)** Fluorescence titration spectra of EhGEF protein with different molar ratio of GTP.**(B)** The fluorescence changes ($\Delta F$) at 330 nm against GTP concentrations.

fluorescence binding study showed significant quenching with increasing concentration of GTP (**Fig 11A**). The $K_D$ value of GTP binding to EhGEF was found to be 298μM (**Fig 11B**). This also indicates that putative GEF domain is sufficient to bind GTP and carry out its biochemical role.

## Discussion

GEF proteins facilitate the activation of Rho-related proteins (small GTPase) by exchanging the GDP with GTP. Switching between nucleotides regulates the inactive or active state of Rho GTPases. Direct activation of small GTPases with their key regulators (GEF) mediates many critical cellular functions in high eukaryotic cells, such as regulation of cell morphology, actin dynamics, gene transcription, cell cycle progression, apoptosis, and tumour progression in human cell [64]. Amoebic trophozoites show proteolytic activity, adherence, phagocytosis, metastatic and invasive behaviour and some of these characteristics are similar to phenotype of metastatic cells [65,66]. During invasive and metastasis processes, trophozoites interact with different host cells and tissues which results in signalling cascades initiating endocytic processes like, phagocytosis and trogocytosis along with motility. During all the before mentioned processes, the small GTPases play important role as they are master regulators for cytoskeletal dynamics and following processes. The cell biology of these GTPases and their regulators remains poorly worked out in this protozoan parasite. Here we discuss findings about a nonconventional and non-Dbl GEF of EhRho1 of this parasite for first time.

Although our understanding of molecular mechanisms of phagocytosis in *E. histolytica* has progressed significantly in the last two decades[35,51,58], however, there is still insufficient information regarding signalling pathways involved in regulation, and progression through different stages of phagocytosis in *E. histolytica*. A number of components are required for formation of functional phagocytic complex and regulating the growth of directional actin network at the particle attachment sites. Members of the Rho family GTPases are known to play central roles in actin dynamics through modulating the activity of actin-regulating proteins, where GEFs work as upstream regulator of Rho proteins activity [38,67].

Although biophysical aspect of EhRho1 have been worked out in detail, [29,31,68] but, its cell biological relevance came out recently [30,35], where EhRho1 is shown to be involved in blebbing and regulates actin cytoskeleton through EhFormin1 and EhProfilin1[69]. We resorted to affinity based enrichment followed by mass spectrometry to decipher the Rho1 mediated signalling pathways. In mass spectrometric screening, EhGEF (EHI_008090) came with high confidence score among the EhRho1 interacting proteins. The peptide sequence coded for well conserved PH domain on N-terminal but rest of the sequence could not be assigned known domain or motif. Our structural bioinformatics analysis of the EhGEF revealed it to be a PT-barrel structure and was also capable of binding GTP and not similar to any Dbl-family GEF structure. This finding came with high confidence, and we were intrigued to test EHI_008090 as a GEF of EhRho1.

Sequence analysis of the peptide revealed that the EhGEF is distinct from other GEF family in length as well as sequence-wise. Secondary structural prediction from different softwares found that EhGEF is consisting substantial portion of α helices and β-sheets where the conventional Dbl-family GEF structures are exclusively formed by α helices. The same was confirmed by CD spectroscopy experiments that alpha helices and beta sheets were 31% and 21% respectively. 3D structure prediction results showed that EhGEF has PT barrel fold and scaffolds similar to prenyltransferases. Further, molecular docking and dynamic simulations of GTP-EhGEF complex is stable and GTP is binding with high confidence. Further, the binding affinity of GTP to EhGEF was confirmed by the fluorescence titration spectra with different molar ratio of GTP and $K_D$ found to be 298μM which is moderate binding constant for any ligand & protein. Interaction between the EhGEF and EhRho1 molecules was investigated by immunoprecipitation and pull-down assays where EhGEF was only able to interact with wild type form of EhRho1 but not with constitutively active (Q63L) or dominant negative (T34N) mutants of EhRho1, which also explains no enrichment of T34N mutant to phagocytic sites as observed previously[30]. The key function of GEF is exchanging the bound GDP in Rho protein pocket with GTP. Neither of EhRho1 mutant Q63L and T34N support this exchange due to mutation in their regulatory domain[35]. The biochemical function of EhGEF was confirmed by *in vitro* nucleotide exchange from GDP to GTP. In MANT-GTP binding assay, EhGEF showed nearly no activity towards EhRab1a, while EhRho1 was well activated. However, EhGEF itself displayed slow binding with GTP, which was also confirmed by tryptophan quenching assay as well. But the florescence of MANT-GTP in reaction containing both EhRho1 and EhGEF showed much increased rate and intensity, which indicates that EhGEF may exchange nucleotides on EhRho1 by potentially a new mechanism which is interesting to be identified. Similarly, EhGEF also showed specific exchange activity for EhRho1 in MANT-GDP exchange assay. To further add, Rhotekin bead assay, which specifically confirmed increase in activated form of EhRho1 (GTP bound) in EhGEF overexpressing transfectants as shown in S5A Fig confirms GTP exchange property towards EhRho1. Although, a Dbl-family EhGEF1 previously identified is reported to exchange GTP on EhRho1 *in vitro* but its relevance to the regulation of EhRho1 has not been reported. Moreover, EhGEF1 localises to uroids and its overexpression leads to decrease in F-actin content. Thus, the previous findings reported might be due to *in vitro* reaction conditions which are different from intracellular environment. Our findings suggest, EhGEF to be one of the specific key activator and regulator of EhRho1 *in vivo*. Sequence analysis suggests that EhRho1 is a homologue of human HsRhoA though the amoebic molecule lacks one of the characteristic and conserved features of Rho GTPases, the "Rho insert domain," required for activation of downstream Rho-associated kinase. However, functionally both human and amoebic molecules appear to be similar as overexpression of EhRho1 was able to generate stress fibres in mammalian [68,70]. EhRho1 reside in cytosol in GDP-bound state and active form of EhRho1 (GTP-bound) translocate

towards membrane and attached to membrane by inserting prenylation domain. Like EhRho1, the localization of EhGEF is also observed in cytosol and membrane of trophozoites. EhGEF can bind to PtdIns(3,4,5)P3 present in the membrane by its PH domain and activate EhRho1 in a location-specific manner. Before the nucleotide exchange, GEF first interact with Rho-GDI complex to release the bound GDI which expose prenylation domain and could initiate the translocation of Rho-GEF complex [71,72]. When the expression of EhGEF was downregulated, recruitment of EhRho1 at phagocytic site was hampered, thus, it can be speculated that EhGEF recruits EhRho1 to the site.

In other higher eukaryotic systems, involvement Rho-GEF in phagocytosis has been well studied. Rho-GEF complex has been found with Fc receptors during phagocytosis [73,74]. So we have also examined the participation of EhGEF in phagocytosis. Two main lines of evidence suggest that EhGEF is involved in phagocytosis; its presence at phagocytic cups along with EhRho1 and reduction in the formation of phagocytic cups on its down-regulation of expression. It has been already established that a threshold concentration of active EhRho1 (GTP bound) is required at the RBC attachment site for initiation and progression of the phagocytic cup. EhGEF may play an important role in achieving this threshold number of active EhRho1 for initiation of phagocytosis. We have observed that phagocytosis continues to take place at a slower pace, in trophozoites down regulated for EhGEF expression, which can be explained, as some amount of EhGEF is present even on down-regulation by antisense RNA, which prolongs the time taken for active EhRho1 to reach the critical level at the site. Conversely, over-expression of EhGEF resulted in an increase in phagocytic events. Thus, the rate of formation and progression of phagocytic cups may be directly proportional to the concentration of some key molecules, such as EhRho1 and its regulators at the phagocytic cups. This observation is in good agreement with the previous reports where siRNA-mediated silencing of HsRac1-GEFs, CrkII or DOCK-180 was sufficient to decrease phagocytosis of IgG-opsonized beads. This decrease of phagocytosis was caused by failure of cells to recruit HsRac1 to the phagocytic site when one of three protein is absent [75]. Similar reports have been reported for other Rho-GEFs in other systems [76–78].

Our microscopy data with trophozoites incubated with RBC clearly show that EhGEF and EhRho1 enrich at the phagocytic cups. The enrichment of EhGEF was observed in the tips of the phagocytic cups which co-localised with EhRho1 and other molecules known to be involved in phagocytosis. The live cell imaging data with GFP-EhGEF further showed the localization of the protein to the plasma membrane site in contact with the RBC or dead CHO cells. The dynamics of the GFP-EhGEF followed a zipper like pattern and progressed as the plasma membrane made contact with target cell to engulf. The GFP-EhGEF disappeared soon after the closure of the phagosome, which was also seen in case of fixed trophozoites. Further, the live imaging of trogocytosis of CHO cells showed recruitment of GFP-EhGEF to tongs like structures formed by plasma membrane in contact with live CHO cells. In this case also, enrichment of molecules in tong like structures was lost once process ceased. The schematic cartoon given in **Fig 12**, shows overall interplay of molecules during an actin dependent endocytic process. The interaction of receptor on amoebic surface with ligands on target cells initiates PtdIns(3,4,5)P3 generation in inner leaflet of plasma membrane, which causes recruitment of EhGEF through PH domain. The localized recruitment of EhGEF leads to spatially restricted activation of EhRho1 at phagocytic site and results in directional action polymerization. As the contact between target cell and amoebic plasma membrane progresses the EhGEF also recruits in PtdIns(3,4,5)P3 dependent manner and further results in actin polymerization which provides necessary force to plasma membrane for endocytosing the target cell or nibbling it. Following closure of plasma membrane, the EhGEF leaves the phagocytic site immediately leaving behind actin, Rho and EhProfilin1. The dynamic nature of EhGEF

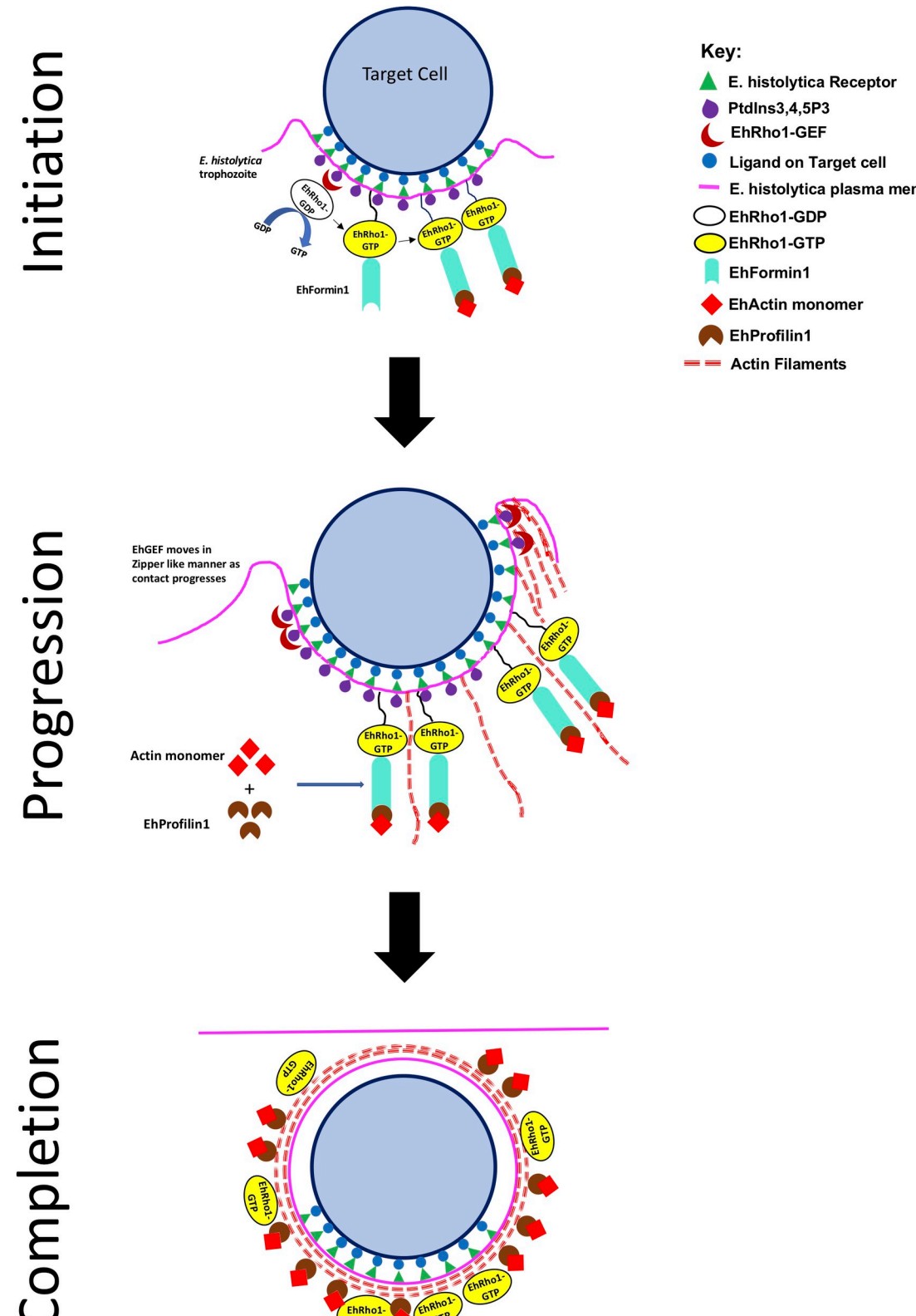

**Fig 12. Schematic cartoon showing role of EhGEF in actin dependent endocytic process of *E. histolytica*.** The interaction between amoebic receptors and ligand on host cells illicit PtdIns(3,4,5)P3 formation on innerleaflet of plasma membrane. The

generation of PtdIns(3,4,5)P3 leads to recruitment of EhGEF along with EhRho1, which regulates actin dynamics through EhFormin1 and EhProfilin1. As the contact between trophozoite and host cell progresses, EhGEF is also recruited to the new site, following a zipper like mechanism. EhFormin1 leads to formation of new actin filaments while EhProfilin1 leads to incorporation of actin monomers to the growing filaments. The growing actin filaments push the plasma membrane to engulf or bite the host cell. Following ingestion, the newly formed endosome is coated with actin, EhRho1 and EhProfilin1 while EhGEF leaves the site immediately.

must be related to the fast turnover of the phosphoinositides at the phagocytic sites which are rapidly converted by action of phosphatases [79] and might be the reason of not being captured in any phagosome proteome studies. Hence, our screening has revealed another important dynamic molecule involved in actin dependent endocytic processes of *E. histolytica*. Also, it is very evident from images that EhGEF downregulation affect the recruitment of EhRho1 and subsequently, EhFormin1 and EhProfilin1 in phagocytic cups, which is similar to previous observation in case of EhRho1 T34N mutant [35]. Result from Rhotekin-RBD pulldown assay also supports the outcome of imaging experiment where EhRho1 was absent in phagocytic cups in EhGEF knockdown cells. Hence, it cannot be ruled out that EhGEF might be involved in targeting EhRho1 to phagocytic site in PtdIns(3,4,5)P3 dependent manner.

In conclusion, we identified EhGEF as EhRho1 interacting protein which regulates its activity during *E. histolytica* phagocytosis. The EhGEF is a unique non-DBl family Rho-GEF, which is capable of exchanging nucleotide on EhRho1 and hence regulate actin dynamics through EhFormin1 and EhProfilin1. The mechanism of recruitment of EhRho1 and EhGEF to the endocytic sites requires more experiments, but our findings are here to initiate the work. As evident from the results, blocking phagocytosis inhibits nutrition of amoeba, EhGFE and other molecules in this pathway can provide new targets for therapeutic intervention in treatment of amoebiasis, when higher drug tolerance to metronidazole has been reported [80]. Lastly, *E. histolytica*, being an early branching eukaryote, deciphering molecular mechanism of phagocytosis in this organism, will help us to understand the evolution of related pathways and molecules in metazoan systems via unique molecules.

## Methods

### Ethics statement

Both mice and rabbits used for generation of antibodies against EhRho1 were approved by the Institutional Animal Ethics Committee (IAEC), Jawaharlal Nehru University (IAEC Code No.: 18/2010). All animal experimentations were performed according to the National Regulatory Guidelines issued by CPSEA (Committee for the Purpose of Supervision of Experiments on Animals), Ministry of Environment and Forest, Govt. of India.

### Cell culture, maintenance, and stable transfection of E. histolytica

*Entamoeba histolytica* HM1: IMSS trophozoites were cultured and maintained in TYI-S-33 medium supplemented with 125 μl of 250 unit/ml penicillin G (potassium salt from Sigma) and 0.25 mg/ml streptomycin per 100 ml of medium as described before[58]. Log phase cells were transfected and selected for stable transformants by method as described before for *E. histolytica*[58]. Cell lines transfected with tetracycline-inducible vectors and GFP or HA tag vectors were maintained in medium containing 10μg/ml of hygromycin B and 10μg/ml of G418, respectively. For experiments purpose, transformants were grown for 24 hr in respected drugs, and then protein expression was induced by adding 30μg/ml tetracycline in tetracycline-inducible system and 30μg/ml of G418 for GFP/HA expression system respectively for 48 hr.

## Cloning of various constructs used in this study

EhGEF was cloned in amoebic expression vector pEhHYG-tetR-O- CAT by replacing CAT gene with the gene of interest in the cassette using double digestion by KpnI and BamHI either in sense or antisense orientations. EhGEF was cloned into pEh-Neo-GFP vector under Xho1 and BamH1 sites so that GFP comes at N-terminus of protein. HA-tagged GEF was cloned in pEh-Neo-3HA vector under SmaI and XhoI sites. The ΔEhGEF was generated by introducing stop codon at 121 position (Alanine to stop codon) by site directed mutagenesis.

## Immunoprecipitation

Cell lysate for immunoprecipitation contained 10 mM Tris-HCl, pH 7.5, 150 mM NaCl, 2 mM p-hydroxymercuribenzoic acid (PHMB), 1 mM phenylmethylsulfonyl fluoride (PMSF), protease inhibitor cocktail, 2 mM β-ME and 1% Triton ×100 and was prepared as described before [58]. It was used after centrifugation at 15,000 rpm for removing cellular debris. Anti-EhRho1 antibody was conjugated to CNBr-activated Sepharose (1 g, Pharmacia) that was activated and processed as per the manufacturer's protocol. The conjugated CNBr-Sepharose beads were incubated with *E. histolytica* lysate (500μg) for 4h at 4˚C. The beads were then washed with wash buffer (10 mM Tris-Cl (pH 7.5), 150 mM NaCl, 1 mM imidazole, 1 mM magnesium acetate, 2 mM β-ME, 0.1% Triton ×100 and protease inhibitor cocktail) thrice. Beads were washed with 0.06 mM Tris-Cl (pH 6.8) and 100 mM NaCl and finally with 0.06 mM Tris-Cl (pH 6.8). The bound proteins were eluted with 0.2M glycine at pH2.0 and eluent was immediately neutralized by 1M tris pH8.0. The sample was then analysed by mass spectrometry.

## Lipid overlay assay

The total *E. histolytica* cells were lysed in the presence of general phosphatase inhibitors like sodium orthovanadate, ß glycerophosphate and sodium fluoride. The lysate was ultracentrifuged at 100,000×g for 1h at 4˚C and supernatant obtained was used for incubation with the membranes spotted with different phospholipids (Echelon) at 4˚C for 1 h. After extensive washing, the membranes were probed with anti-HA antibodies, followed by secondary HRP-labelled anti-mouse IgG antibodies (Invitrogen).

## Assay for destruction of CHO monolayer

The destruction of CHO monolayers was quantified as described previously with slight modifications. Briefly, CHO cells were labelled with 40 mM of CellTracker blue in the growth medium at 37˚C for 2 h. After the medium was replaced with pre-warmed OPTI-MEM (Invitrogen-Gibco) supplemented with 137 mM L-cysteine and 19 mM ascorbic acid, pH 6.7, ~$1.0 \times 10^5$ *E. histolytica* trophozoites were added and incubated at 35˚C for 60 min. After this incubation, the dish was washed with cold medium to remove trophozoites and the remaining CHO cells were collected using trypsin and the fluorescence of CellTracker blue was measured using a fluorometer (F-2500, Hitachi, Japan) with excitation and emission at 353 and 465 nm, respectively. The number of adherent CHO was proportional to the intensity of CellTracker blue staining and expressed as a percentage of the remaining fluorescence of untreated CHO/Caco-2 cells.

## Amoebic cell extract and sub cellular fractionation

The $1 \times 10^5$ amoebic cells were harvested in log phase by centrifugation at 280 g for 7 min at 4˚C and washed with ice cold PBS. Cells were lysed in Lysis buffer (10 mM Tris-Cl pH 7.5, 150 mM NaCl, 1% Triton-X100, 2 mM Polyhexamethylene Biguanide (PHMB), and 1× protease

inhibitor cocktail [Sigma]) and centrifuged at 13,000 g for 20 min to pellet down the debris. The supernatant was collected and labelled as total cell lysate. To separate membrane from cytosol fraction, subcellular fractionation was done. Briefly, $10^7$ trophozoites growing in log phase were harvested at 600 g for 5 min at 4˚C. The cell pellet was washed with 1× PBS pH 8.0. The cells were resuspended in lysis buffer (10 mM Hepes pH 7.5, 1.5 mm MgCl2, 10 mM KCl, 5 mM DTT, 0.2% NP-40, 1× PIC, 2 mM PHMB, and 2 mM phenylmethylsulfonyl fluoride (PMSF)) and incubated in ice for 15 min followed by centrifugation at 3,000 g for 10 min at 4˚C. The supernatant and pellet were separated. The nuclear fraction was obtained by resuspension of pellet in lysis buffer. The supernatant was further subjected to high-speed ultracentrifugation at 100,000 g for 30 min. at 4˚C. The supernatant obtained was labelled as cytosolic fraction, and the pellet was labelled as membrane fraction. The pellet obtained post ultracentrifugation was washed with 1× PBS pH 8.0 and resuspended in membrane solubilizing buffer containing 100 mM Tris pH 7.5 and 1% Triton X-100.

### Immunostaining of *E. histolytica* cells

Immunofluorescence staining of *E. histolytica* cells was carried as described before[58]. In brief, log phage *E. histolytica* cells were harvested, washed with phosphosaline buffer (PBS) #8 and resuspended in incomplete TYI-33 medium. Cells were allowed to adhere on acetone-precleaned coverslips in 35-mm petri dishes at 37˚C for 10 min. After that, cells were fixed in 3.7% prewarmed paraformaldehyde for 30 min followed by permeabilisation with 0.1% Triton X-100/PBS for 5 min at 37˚C. After wash with PBS, cells were quenched for 30 min in PBS containing 50-mM NH$_4$Cl. Blocking has been done by 1% BSA/PBS for 30 min, followed by incubation with primary antibody at 37˚C for 1 hr. Coverslip was washed three times with 1% BSA/PBS and incubated with secondary antibody for 30 min at 37˚C. Antibody dilutions used were as follows: anti-HA at 1:100, anti-EhRho1 at 1:50, anti-EhCaBP1 at 1:200, anti-EhFormin1 at 1:50, anti-EhProfilin1 at 1:50, 1:300 dilution of anti-rabbit Alexa 488, 556 and anti-mice Alexa556 (Molecular Probes), and Tetramethylrhodamine (TRITC)-Phalloidin was used at 1:250. Coverslip was washed with PBS for three times and mounted on a glass slide using DABCO (1,4-diazbicyclo (2,2,2) octane (Sigma) 2.5% in 80% glycerol). The edges of the coverslips were sealed with nail-paint to avoid drying. Confocal images were visualized using an Olympus FLUOVIEW FV1000 laser scanning microscope with objective lenses PLAPON 60× O, NA- 1.42, and Nikon. The raw images were processed using FV10- ASW 1.7 viewer and NIS element 3.20. Colocalisation analysis was done by using NIS element 3.20 or FV10-ASW 1.7. A constant area was chosen as a ROI throughout the analysis. We have calculated the relative intensity by using NIS-Elements AR 3.0 within this ROI and intensity were plotted as 100% for each marker separately.

### Fluorescent labelling of RBCs

RBCs were stained with CFSE (Carboxyfluorescein succinimidyl ester) following a modified protocol (Cell Trace CFSE proliferation kit, Invitrogen). Cells ($2\times10^7$ cells/ml) were incubated in CFSE staining buffer (PBS containing 0.1% BSA and 2.5 µM CFSE) for 10 min at 37˚C. The reaction was blocked with complete medium in presence of 2% serum for 10 min on ice, after which, RBC were washed three times with an excess of incomplete media of *E. histolytica*.

### Live cell imaging

Approximately $5 \times 10^5$ transformants were cultured on a 35 mm collagen-coated glass-bottom culture dish (MatTek Corporation, Ashland, MA) in 3 ml of TYI-S-33 medium under anaerobic conditions. CHO cells were stained for 30 min with 20 mM CellTracker orange dye

(Molecular probes, Eugene, OR) in F12 medium containing 10% FCS. After staining, CHO cells were washed three times with fresh F12 medium, and ~2 × 10$^5$ CHO cells in 200 μl F12 medium were added to the GFP-tagged protein-expressing amoeba in a glass-bottom dish. The culture was carefully covered with a coverslip, and over- loaded medium was removed. The junction of the coverslip and slide glass was sealed with nail polish, and the culture was incubated at 35˚C in a temperature control unit on Zeiss, LSM780 equipped with a ×63/1.4 oil immersion objective and CCD camera.

## Erythrophagocytosis assay

Briefly, 10$^6$ RBCs were washed with PBS and incomplete TYI-33 and were then incubated with 10$^5$ *E. histolytica* for varying times at 37˚C in 0.5 ml of culture medium. The trophozoites and erythrocytes were centrifuged to get a pellet, non-engulfed RBCs were lysed with cold distilled water and recentrifuged at 1000×g for 2 min and step was repeated twice, followed by resuspension in 1 ml formic acid to lyse *E. histolytica* containing engulfed RBCs. The absorbance was measured at 400 nm.

## GEF exchange assay

GEF exchange assay was performed as per manufacturer protocol (Cytoskeleton, BK100). In brief, the exchange reaction was set in 100μl volume by adding 50μM respective GTPase, 5μM EhGEF1 and 1X reaction buffer in 96 well plate. Fluorescence reading was collected at λEx = 505 nm, λEm = 515 nm for 15 min and reading was computed using graph pad prism.

## MANT-GDP dissociation assay

MANT-GDP dissociation assay was performed as described before [47]. In brief, MANT-GDP was loaded on respective GTPases in loading buffer (20mM HEPES, 50mM NaCl, 0.5mM MgCl$_2$, 10mM EDTA, 2mM DTT and 10mM MANT-GDP). After MANT-GDP loading, an exchange reaction was set up with 10 fold excess GTP in nucleotide exchange buffer (40mM HEPES, 50mM NaCl, 10mM MgCl$_2$ and 2mM DTT). The fluorescence reading was collected every 15sec. at λEx = 360 nm, λEm = 440 nm and computed using graph pad prism.

## Protein structure modelling

The 3D structures of EhGEF were modelled using various tools including Rosetta (Comparative and *ab initio* modelling), TrRosetta, I-TASSER and SWISS-MODEL [81–83]. The predicted structures were refined using GalaxyRefine webserver[84]. Validation of the predicted structures was done based on metrics provided by independent webservers including Molprobability, QMEAN, ERRAT, Verify 3D and ProSA[85–88]. The best model was selected for molecular docking studies with GTP as ligand.

## Docking and molecular simulations

The binding pocket of the model was predicted using CASTp server and used for selection of grid parameters[89]. Docking was performed using Autodock Vina with an exhaustiveness of 16[90]. The docked complex of EhGEF with GTP along with unliganded EhGEF model were subjected to molecular dynamics simulations using GROMACS software suite (version 2018.7)[63]. GROMOS 54a7 force field was employed for generating topology of the protein while ligand topology was generated from PRODRG server[91]. The complex was centered in a cubic box at least 1nm from the box edge and solvated with SPC water molecules. The system was neutralized by addition of adequate ions. The electroneutral system was energy minimized

using steepest descent algorithm with a force convergence criteria of less than 1000 kJ mol$^{-1}$ nm$^{-1}$. The energy minimized system was subjected to equilibration in two phases of 100ps duration, keeping the protein-ligand complex restrained. The first phase of equilibration was performed under a canonical ensemble (NVT) followed by second phased under an NPT ensemble to stabilize the temperature (300K) and pressure (1bar) of the system. V-rescale, a modified Berendsen thermostat and Parrinello-Rahman barostat was used for temperature and pressure coupling respectively. The bond lengths were constrained using P-LINCS algorithm and long-range electrostatics were computed using Particle Mesh Ewald (PME) scheme [92,93]. The unrestrained production simulation was performed for a duration of 200ns using leap-frog dynamics integrator with a step size of 2fs. Periodic boundary conditions were applied in all three dimensions. Analysis of the simulation trajectory was done using modules available in GROMACS and in-house python scripts. Similar methodology was adopted for simulation of unliganded EhGEF model.

## Circular dichroism spectroscopy

CD spectroscopy was performed to probe the secondary structure of the protein. The CD spectra of EhGEF at 25°C was obtained on a Jasco Spectropolarimeter (Model J-1500). Baseline subtractions were done with CD spectrum signal of buffer containing 50 mM Tris and 150mM NaCl at pH 7.5. The far-UV (200–250 nm) spectra were recorded in triplicates using a quartz cell with a path length of 0.1 cm, a response time of 1 s, a scan speed of 20 nm/min and bandwidth of 0.5 nm. The ellipticity was normalized to a concentration-independent unit mean residue ellipticity $[\theta]_{MRE}$ (deg cm$^2$ dmol$^{-1}$), using the following formula,

$$[\theta]_{MRE} = \frac{MRW \cdot \theta_{\lambda}}{10 \cdot c \cdot l} \tag{1}$$

where $\theta_{\lambda}$ is the observed ellipticity in millidegrees, MRW is the mean residue weight of the protein residues (~110 Daltons), c is concentration in mg ml$^{-1}$ and l is the path length of the cell in cm.

The percentage secondary structure estimation was performed from the CD spectra obtained at 0.4 mg/ml protein concentration using DichroWeb server employing K2D neural network-based algorithm[94]. The percentage secondary structure obtained from CD spectroscopy studies was compared with those estimated from protein sequence and modelled structure using PSIPRED and STRIDE servers respectively[95,96].

## *In vitro* fluorescence binding study to probe binding of GTP with EhGEF

*In vitro* binding interaction of EhGEF with GTP was probed through tryptophan quenching assay. The assay was performed at 25°C in JASCO spectrofluorometer (Model J-1500) with 1cm path length quartz cuvette. Emission spectra of 2μM EhGEF, in 50mM phosphate buffer and 50mM NaCl at pH 7.4, was recorded from 300nm to 450nm after excitation at 295nm. The protein was titrated with an increasing concentration of GTP (0–400 μM) and emission spectra were recorded. The emission and excitation slit width was set at 5nm throughout the experiment. Spectra were recorded with 200μM GTP in assay buffer for baseline correction. Curve fitting was done with non-linear regression using a Gaussian equation for the saturation curve and one-site specific binding Eq (2) for the binding curve,

$$Y = \frac{Bmax * X}{Kd * X} \tag{2}$$

Where Bmax is the maximum specific binding and Kd is the equilibrium dissociation constant (μM).

## Statistical analysis

Statistical comparisons were made using ANOVA test. Experimental values were reported as the means ± standard error. Differences in mean values were considered significant at *p-value≤0.05, **p-value≤0.005, ***p-value≤0.0005. All calculations of statistical significance were made using the GraphPad InStat software package (GraphPad).

## Supporting information

**S1 Fig. Quantitative analysis of subcellular fractionation immunoblots.** Quantitative analysis of immunoblots of subcellular fractionation from Fig 3A using software AlphaEaseFC 4.0 based on three independent experiments.
(TIF)

**S2 Fig. Comparative localization of EhGEF with respect to phagocytic markers.** Co-localization was studied in immunostained images of EhGEF during erythrophagocytosis assay with indicated phagocytic marker proteins during **(A)** Initiation of phagocytic cup formation at (1–3) minutes **(B)** Progression of phagocytic cups at (5–7) minutes and **(C)** Phagosome formation at (7–10) minutes. 48h grown *E. histolytica* cells were incubated with RBC for different time intervals at 37˚C and subsequently fixed for immunostaining. Cells were immunostained with HA-tag specific antibody followed by Alexa-488 conjugated secondary antibody (green), F-actin was stained with TRITC conjugated phalloidin (red). EhCaBP1, EhCaBP3 and EhC2PK were immunostained with protein specific antibodies followed by Alexa-405 conjugated secondary antibody (blue). Arrowheads indicate phagocytic cups with enrichment of indicated proteins. **(D)** Quantitative analysis of fluorescent signals of indicated proteins in different steps of phagocytosis in immunostained images of *E. histolytica* cells. Four regions were selected from cytosol, phagocytic cups, just closed cups and phagosomes for each cells. Average intensity was calculated for each region. Relative intensities were calculated by assuming intensity as 100% for each marker separately. This experiment was carried out by selecting randomly five cells in triplicates. (N = 5, bar represent standard error). Bar represent 10μm, DIC is differential interference contrast. **(F)** Co-localization coefficient was analyzed from 10 cells using NIS 4.0 AR software. PCC(r) value of EhGEF with EhCaBP1, EhC2PK and EhCaBP3 during phagocytic cup formation is indicated. ANOVA test was used for statistical comparisons.*p-value≤0.05, **p-value≤0.005, ***p-value≤0.0005.
(TIF)

**S3 Fig. GFP-EhGEF expression and localization in amoebic cells during phagocytosis. (A)** Western analysis of GFP -EhGEF. *E. histolytica* cell expressing GFP-EhGEF were induced with different concentration of G418(10, 20 and 30 μg/ml) for 48h. Each lane was loaded with 100 μg of cell lysate as shown in figure. Blot probe with anti-GFP antibody and EhCoactosin was taken as a loading control for experiment. **(B)** Fluorescence images of *E. histolytica* cells expressing GFP-EhGEF during phagocytosis of RBC (Red). Cell were induced as mention above and immunostained with anti GFP tag specific antibodies followed by Alexa 488 and F-actin was stained with TRITC phalloidin.
(TIF)

**S4 Fig. EhGEF expressing cell line. (A)** Schematic representation of *E. histolytica* specific tetracycline inducible pEhHyg-TetR-O-CAT (TOC) vector. EhRho1 was cloned in sense and antisense orientation in BamH1 and Kpn1 sites of pEhHyg-TetR-O-CAT vector. **(B)**

Proliferation of *E. histolytica* trophozoites carrying different constructs was studied. All cells were grown in presence of 10 μg/ml hygromycin and tetracycline was added to the medium at 30 μg/ml at starting time. Cells were grown in 5 ml culture tubes in triplicate for all the experiments and counting was carried out using a haemocytometer, after chilling the tube for 5 min. One-way ANOVA test was used for statistical comparisons.
(TIF)

**S5 Fig. EhGEF regulates the localization of molecules important for phagocytosis. (A)** Rhotekin GST-Rho-binding domain (RBD) pull-down assay. Glutathione sepharose beads bound to GST-tagged Rho-binding domain of Rhotekin was incubated with indicated cell lysate for 4 hr during which the activated EhRho1 binds to GST-RBD. The beads were washed three times before analysis by western blotting. For internal and loading control, 100 μg of lysate was taken prior to incubation of beads with the lysate. The activated EhRho1 decreases markedly in EhGEF antisense cells. **(B)** Immunofluorescence images of *E. histolytica* cells expressing anti-sense EhGEF during erythrophagocytosis. Cells were stained with anti-EhRho1, anti-EhFormin1 or anti-EhProfilin1 antibodies followed by Alexa-405 or Alexa-488 secondary antibodies. Actin was stain with TRITC-phalloidin. Bar represent 10μm, DIC is differential interference contrast. **(C)** Western blot analysis of amoebic cells expressing indicated constructs showing the level of EhRho1 in vector alone, antisense (AS) and sense EhGEF in the presence and the absence of tetracycline. EhCoactosin1 was used as an internal and loading control.
(TIF)

**S6 Fig. EhGEF-GTPcomplex RMSD profile. (A)** The equilibrated RMSD profile shows no significant conformational changes in the modelled EhGEF and also the EhGEF-GTPcomplex. **(B)** The radius of gyration showing the compactness of EhGEF and EhGEF-GTPcomplex.
(TIF)

**S7 Fig. CD spectroscopy for recombinant EhGEFΔPH molecule.** The CD spectra confirms the α/β content of secondary structure as predicted from the sequence information and modelled structure.
(TIF)

**S1 Table. Table listing the proteins identified as EhRho1 and EhGEF binding proteins respectively in affinity screening by mass spectrometry.** The proteins are arranged in order of sequence coverage in the mass spectrometry data.
(DOCX)

**S1 Movie. The CFSE labeled human erythrocytes (red) were engulfed by GFP-EhGEF expressing trophozoite.** The GFP-EhGEF enriched on the membrane in contact with RBC and followed a zipper like movement as contact progressed.
(AVI)

**S2 Movie.** Amoebic trophozites (green) expressing GFP-EhGEF phagocytosed the heat killed CHO cells (red). The dead host cells are ingested via phagocytosis in which the involvement of EhGEF can be seen in a similar manner as S1 Movie.
(AVI)

**S3 Movie. GFP-EhGEF expressing trophozoites engulfing the red colored live CHO cells via trogocytosis.** The trophozites ingest live cells in small bites and GFP-EhGEF can be seen to enrich on the tips of the tong like apparatus formed by the parasite.
(AVI)

## Author Contributions

**Conceptualization:** Somlata.

**Data curation:** Ravi Bharadwaj, Krishna K. Inampudi, Somlata.

**Formal analysis:** Ravi Bharadwaj, Krishna K. Inampudi, Somlata.

**Funding acquisition:** Tomoyoshi Nozaki, Somlata.

**Investigation:** Ravi Bharadwaj, Tushar Kushwaha, Azhar Ahmad, Krishna K. Inampudi, Somlata.

**Methodology:** Ravi Bharadwaj, Tushar Kushwaha, Azhar Ahmad, Krishna K. Inampudi, Somlata.

**Project administration:** Tomoyoshi Nozaki, Somlata.

**Resources:** Krishna K. Inampudi, Tomoyoshi Nozaki, Somlata.

**Supervision:** Somlata.

**Visualization:** Ravi Bharadwaj, Somlata.

**Writing – original draft:** Ravi Bharadwaj, Somlata.

**Writing – review & editing:** Ravi Bharadwaj, Azhar Ahmad, Krishna K. Inampudi, Tomoyoshi Nozaki, Somlata.

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
