## [Decision Letter · Decision Letter 0]

13 Oct 2021

Dear Dr Somlata,

We are pleased to inform you that your manuscript 'An atypical EhGEF regulates phagocytosis in Entamoeba histolytica through EhRho1' has been provisionally accepted for publication in PLOS Pathogens.

Best regards,

William A. Petri, Jr.

Associate Editor

PLOS Pathogens

Vern Carruthers

Section Editor

PLOS Pathogens

Kasturi Haldar

Editor-in-Chief

PLOS Pathogens

orcid.org/0000-0001-5065-158X

Michael Malim

Editor-in-Chief

PLOS Pathogens

orcid.org/0000-0002-7699-2064

Reviewer Comments (if any, and for reference):

Reviewer's Responses to Questions

**Part I - Summary**

Reviewer #1: Given the importance of EhRho1 in phagocytic processes towards phagosomes formation, the authors tried identifying the molecules interacting with EhRho1 by immunoprecipitation and mass spectrometry analysis. The study showed several novel proteins interacting with EhRho1 with a high confidence score, including a Guanine Nucleotide Exchange Factor (EhGEF, EHI_008090). Then, vice versa, the molecules interacting with EhGEF were also identified, and within them, EhRho1 also interacted with a high confidence score.

During the progression of cups until the closure of phagosomes, EhGEF was found to localize in the phagocytic cups formed on E. histolytica but not in the phagosomes themselves. The authors showed that EhGEF, mainly a cytoplasmic protein, was recruited to the site of phagocytosis and trogocytosis in response to phosphatidylinositol 3,4,5 triphosphate (PtdInsP3) transiently and also recruited EhRho1 to the site. The authors identified EhGEF, a unique non-Dbl Guanine Nucleotide Exchange Factor, to interact with EhRho1 and regulate its activity by exchanging nucleotide on EhRho1 via an unconventional pathway during E. histolytica phagocytosis. Thus, EhGEF can control actin dynamics through EhFormin1 and EhProfilin1 and hence is required to initiate phagocytosis toward phagosome formation.

This study identified an atypical EhGEF that activates EhRho1 as one of the specific vital activators and regulates phagocytosis in Entamoeba histolytica, which are novel and significant findings of the study. The study is well planned and executed with enough scholarship in this scientific field. If you ask me the weaknesses of the study, it may be the vagueness of relative importance of this EhGEF (EHI_008090) among all of RhoGEFs, including Dbl- RhoGEFs, in E. histolytica.

**Part II – Major Issues: Key Experiments Required for Acceptance**

Reviewer #1: (No Response)

**Part III – Minor Issues: Editorial and Data Presentation Modifications**

Reviewer #1: (No Response)

PLOS authors have the option to publish the peer review history of their article (what does this mean?). If published, this will include your full peer review and any attached files.

Reviewer #1: No

---

## [Editor Report · Acceptance letter]

11 Nov 2021

Dear Dr Somlata,

We are delighted to inform you that your manuscript, "An atypical EhGEF regulates phagocytosis in Entamoeba histolytica through EhRho1," has been formally accepted for publication in PLOS Pathogens.

Best regards,

Kasturi Haldar

Editor-in-Chief

PLOS Pathogens

orcid.org/0000-0001-5065-158X

Michael Malim

Editor-in-Chief

PLOS Pathogens

orcid.org/0000-0002-7699-2064